# LOOK TWICE BEFORE YOU ANSWER: MEMORY-SPACE VISUAL RETRACING FOR HALLUCINATION MITIGATION IN MULTIMODAL LARGE LANGUAGE MODELS

## ABSTRACT

Despite their impressive capabilities, Multimodal Large Language Models (MLLMs) are susceptible to hallucinations, especially assertively fabricating content not present in the visual inputs. To address the aforementioned challenge, we follow a common cognitive process - *when one's initial memory of critical on-sight details fades, it is intuitive to look at them a second time to seek a factual and accurate answer.* Therefore, we introduce **Mem**ory-space **V**isual **R**etracing (**MEMVR**), a novel hallucination mitigation paradigm that without the need for external knowledge retrieval or additional fine-tuning. In particular, we treat visual tokens as supplementary evidence to be reinjected into MLLMs via Feed Forward Network (FFN) as "key-value memory" at the middle trigger layer, *i.e.*, when the model is uncertain about visual memories in the layer. Comprehensive experimental evaluations demonstrate that MEMVR significantly mitigates hallucination issues across various MLLMs and excels in general benchmarks without incurring added time overhead, thus emphasizing its potential for widespread applicability. [1]

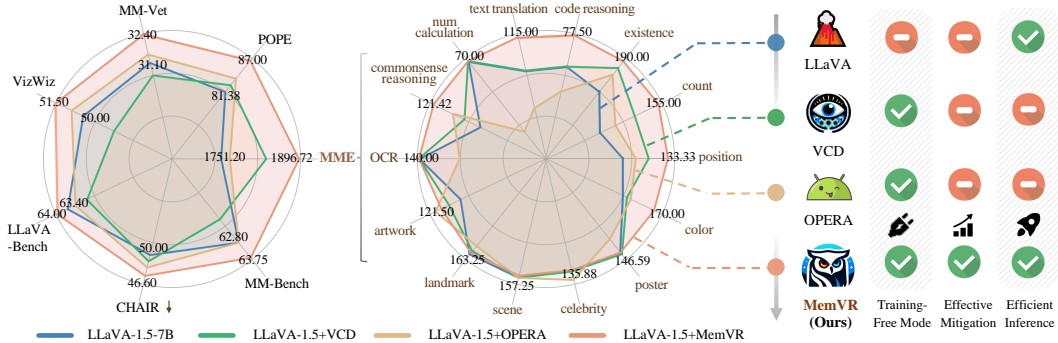

**Figure 1:** **MEMVR** demonstrates strong performance across seven benchmarks spanning various domains (**Left**), with particularly outstanding results on the MME benchmark (**Center**). Additionally, MEMVR is compared with contrastive decoding schemes, standing out for its ability to alleviate hallucinations using just a single inference, making it a more eco-friendly solution (**Right**).

*A moment's insight is sometimes worth a lifetime's experience.* —— Holmes Jr.

## 1 INTRODUCTION

Multimodal Large Language Models (MLLMs), due to their formidable capacity to comprehend visual inputs, have emerged as indispensable tools in computer vision (Koh et al., 2024) and natural language processing (Tu et al., 2023) to tackle numerous visual tasks and facilitate complex visual question answering (VQA). Nevertheless, MLLMs still exhibit certain limitations, *i.e.*, the so-called "hallucination" phenomenon (Huang et al., 2024b; Zheng et al., 2024). Specifically, MLLMs frequently generate descriptions inconsistent with user-provided visual inputs, such as incorrectly outputting non-existent things in the image or conflicting judgments. This flaw poses significant risks to the reliability of MLLMs as trustworthy assistants (Yu et al., 2024b), particularly in safety-critical applications (Zou et al., 2023) (*e.g.*, clinical healthcare (Lin et al., 2024) and autonomous

---

[1]The source code will be available to the public.

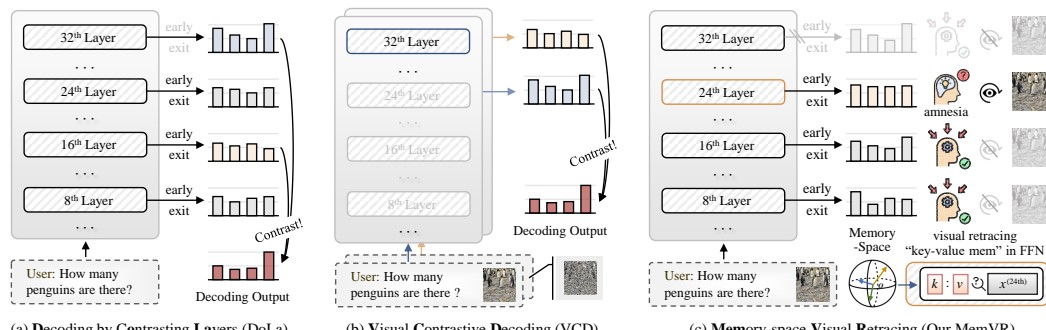

(a) **D**ecoding by **C**ontrasting **La**yers (DoLa)  (b) **V**isual **C**ontrastive **D**ecoding (VCD)  (c) **Mem**ory-space **V**isual **R**etracing (Our MemVR)

Figure 2: The conventional paradigm for hallucination mitigation, employing contrastive decoding methods such as DoLa (a) and VCD (b), is compared to our proposed MEMVR strategy (c).

driving scenarios (Ding et al., 2024)). While the exact reasons for hallucinations in MLLMs are not fully understood, one possible factor could be the imbalance between their understanding of visual and textual information. Typically, MLLMs encode images into vision tokens via CLIP (Radford et al., 2021), which are fed along with text tokens into LLMs for decoding. LLMs excel at text comprehension but struggle with visual information perception and memory, where differences in information density between modalities may cause inconsistencies during decoding.

Numerous methods have been proposed to mitigate hallucinations in MLLMs, and general studies can be broadly categorized into three streams: (i) Retrieval-Augmented Generation (RAG) (Shuster et al., 2021; Caffagni et al., 2024) that incorporates knowledge from external databases to mitigate the problem of "hallucination", as well as (ii) through extra Fine-tuning (Yu et al., 2024b) to enhance the self-consistency of generation, and (iii) Contrastive Decoding (CD) strategy, which do not involve extra training. Specifically, RAG and fine-tuning patterns typically employ external knowledge retrieval or robust instruction-tuning datasets to post-hoc debias (Yang et al., 2024; Liu et al., 2023a), which inevitably introduces substantial computational overhead or storage requirements. For example, some approaches(Yin et al., 2023; Yu et al., 2024a) have fine-tuned models using high-quality visual instructions generated by advanced automated annotation tools including GPT-4 (OpenAI, 2023).

CD-based methods (Li et al., 2023a; Shi et al., 2024) represent a simpler and more efficient way to mitigate hallucinations than RAG and fine-tuning based methods. Particularly, CD-based hallucination mitigation methods usually modulate the logits of the next token prediction through contrast manner or penalty mechanisms. As illustrated in Figure 2 (a), DoLa (Chuang et al., 2023) enriches factual knowledge via layer-wise contrasting and reduces the generation of incorrect facts in LLMs. In MLLMs, VCD (Leng et al., 2024) amplifies the language priors by adding Gaussian noise to the visual inputs, thereby reducing over-reliance on statistical biases and single-modal priors through contrasting output distributions from original and distorted visual inputs as in Figure 2 (b). This perturbation of original inputs requires task-specific design, inevitably **doubling inference costs**. More critically, contrastive distributions are agnostic to visual and instructional nuances, which may not always amplify the intended hallucinations, occasionally **introducing potential noise into CD**.

In this work, we delve into the challenges of hallucination mitigation in MLLMs and address the shortcomings of CD-based approaches. Our research is grounded in a common cognitive process: *when the initial memory of certain critical visual details fades, it is intuitive to look at them for the second time to search for the accurate answer* (O'regan, 1992; Ballard et al., 1995; Horowitz & Wolfe, 1998). Different from visual contrastive decoding strategies that **alleviate hallucinations by diminishing language priors** (Leng et al., 2024; Qu et al., 2024; Park et al., 2024), we propose a novel **Mem**ory-space **V**isual **R**etracing (**MEMVR**) method that mitigates hallucinations through **supplementing visual evidence**, akin to the two sides of a coin. MEMVR, as shown in Figure 2 (c), is an architecture-agnostic, plug-and-play solution that re-injects visual features into an intermediate layer suffering from vision-related memory lapse with only one regular inference. This novel hallucination mitigation paradigm in terms of efficiency, and its inference cost and performance are optimal compared to previous studies as listed Table 1. It's a game-changer for effectiveness

| Method | 20-Token Len | 50-Token Len | 80-Token Len |
|---|---|---|---|
| LLaVA-1.5 | 1880.3 ↓×1.0 | 3617.6 ↓×1.0 | 5256.6 ↓×1.0 |
| + VCD | 4537.4 ↑×2.4 | 7690.8 ↑×2.1 | 11569.3 ↑×2.2 |
| + OPERA | 6242.7 ↑×3.3 | 12672.3 ↑×3.5 | 19247.2 ↑×3.7 |
| + MEMVR | **1861.7** ↑×**1.0** | **4000.9** ↑×**1.1** | **5545.5** ↑×**1.1** |

Table 1: Efficiency Comparisons for generating different length tokens, using an NVIDIA-A40 GPU. Inference time (ms) of different methods is recorded.

and efficiency. Through extensive experiments on multimodal hallucination benchmarks, as well as GPT-4o[2] evaluations, we show the comprehensive performance improvements of MEMVR in hallucination mitigation and general capabilities. Our contributions can be summarized as follows:

❶ We propose MEMVR, a novel training-free hallucination mitigation paradigm that effectively alleviates hallucinations in MLLMs. In contrast to previous methods, which primarily focus on eliminating biases of language priors, MEMVR seeks to replenish question-relevant visual clues towards more evidential responses, which signifies the other side of the coin.

❷ We design a dynamic premature layer injection strategy with visual retracing in MLLMs, mimicking human intuitive thinking to revisit image features for self-consistency and credible answers when pivotal memories are scrambled. Furthermore, we theoretically demonstrate that visual retracing can effectively diminish hallucinations from an information-theoretic perspective.

❸ Comprehensive experimental results demonstrate the effectiveness of MEMVR in mitigating hallucinations and enhancing general cognitive and perceptual performance, as well as its high efficiency. Our research will make a substantial contribution to trustworthy multimodal intelligence. To the best of our knowledge, we are the first to mitigate hallucinations and improve general performance in MLLMs with only one regular inference, without incurring added time overhead.

## 2 RELATED WORK

**MLLMs and Challenges.** In recent years, MLLMs have made remarkable progress, particularly as they have evolved from the foundations laid by Vision Language Models (VLMs). Early based on BERT-style language decoders (Devlin, 2018), which achieved initial cross-modal integration by combining visual and textual data (Li et al., 2022). Leveraging open-source Large Language Models (LLMs) such as LLaMA families (Touvron et al., 2023), MLLMs (Alayrac et al., 2022; Wu et al., 2024) have demonstrated enhanced adaptability across a range of visual language tasks, leading to a more profound ability to interpret the world. Models like LLaVA (Liu et al., 2024), Qwen-VL (Bai et al., 2023), and GLM4V (Wang et al., 2023) have further advanced this field, enabling users to interact with these agents using both image and text prompts. These models adhere to two critical training phases: pre-training feature alignment and instruction fine-tuning, ensuring they better comprehend the format of instruction inputs (Yin et al., 2024). However, despite their impressive performance in many areas, MLLMs still suffer from hallucination issues. Thus, in this paper, we primarily conducted experiments and analysis on these three representative models.

**Hallucinations in MLLMs.** Before LLMs, hallucination in NLP was mainly seen as generating nonsensical or deviant content. In MLLMs, "hallucination" is defined as the model generating content inconsistent with the provided image. This issue stems significantly from inadequate alignment among modalities. Several methods have been explored to mitigate hallucinations in MLLMs. Early efforts focused on fine-grained modality alignment (Rohrbach et al., 2018) and reducing co-occurrence biases (Kim et al., 2023) in small-scale VLMs, but these approaches struggle to scale with MLLMs. More recent strategies involve hallucination-targeted datasets for fine-tuning (Gunjal et al., 2024), post-hoc revisors (Zhou et al., 2024), and adopting RLHF (Yu et al., 2024b). While effective, these methods are resource-intensive. CD-based approaches (Chuang et al., 2023; Leng et al., 2024) adjust the decoding distribution to mitigate hallucinations, but it does not consistently improve performance. Compared with them, our MEMVR stands as "*a paradigm of effectiveness and efficiency*" in hallucination mitigation, effortlessly enhancing performance without extra training.

## 3 METHODOLOGY

In this section, we first formulate the generation process of MLLMs to facilitate a clearer understanding of our MEMVR. Moreover, we introduce our hypothesis about the causes of hallucinations and discuss visual retracing and its dynamic strategy. Further, we conduct theoretical analysis.

### 3.1 MLLMS GENERATION PROCESS

Typically, MLLMs are composed of a visual encoder $C_v$, a text embedding layer, $L$ number of transformer layers, and an affine layer $\zeta(\cdot)$ which predicts the distribution of the next token, with diverse modalities as inputs, *e.g.*, images and text. Regardless of specific architectural variations, MLLMs commonly employ $C_v$ to extract vision tokens from the raw image and project it into

---

[2]GPT-4o-2024-08-06: https://platform.openai.com/docs/models/gpt-4o.

*i*-th early layer (uncertainty of varying layers)

| token | The | _image | _features | _a | _wooden | _d | ining | _table | _filled | _with | _a | _variety | _of | _food | s | , |
|---|---|---|---|---|---|---|---|---|---|---|---|---|---|---|---|---|
| 32 | 0.064 | 0.049 | 0.734 | 0.037 | 0.656 | 0.156 | 0.000 | 0.016 | 0.706 | 0.009 | 0.473 | 0.622 | 0.001 | 0.708 | 0.801 | 0.643 |
| 30 | 0.148 | 0.006 | 0.520 | 0.039 | 0.337 | 0.284 | 0.000 | 0.003 | 0.739 | 0.001 | 0.373 | 0.556 | 0.001 | 0.327 | 0.481 | 0.893 |
| 28 | 0.663 | 0.033 | 0.355 | 0.223 | 0.345 | 0.042 | 0.001 | 0.005 | 0.775 | 0.003 | 0.451 | 0.428 | 0.001 | 0.103 | 0.415 | 0.737 |
| 26 | 0.847 | 0.067 | 0.316 | 0.615 | 0.030 | 0.018 | 0.200 | 0.008 | 0.598 | 0.020 | 0.149 | 0.447 | 0.019 | 0.069 | 0.266 | 0.785 |
| 24 | 0.830 | 0.028 | 0.063 | 0.368 | 0.035 | 0.012 | 0.174 | 0.008 | 0.758 | 0.066 | 0.524 | 0.103 | 0.058 | 0.076 | 0.328 | 0.638 |
| 22 | 0.860 | 0.045 | 0.077 | 0.407 | 0.040 | 0.026 | 0.140 | 0.016 | 0.801 | 0.319 | 0.748 | 0.035 | 0.480 | 0.474 | 0.317 | 0.853 |
| 20 | 0.916 | 0.120 | 0.360 | 0.354 | 0.247 | 0.152 | 0.573 | 0.031 | 0.155 | 0.273 | 0.172 | 0.099 | 0.746 | 0.439 | 0.357 | 0.829 |
| 18 | 0.879 | 0.462 | 0.947 | 0.691 | 0.273 | 0.210 | 0.771 | 0.046 | 0.251 | 0.203 | 0.871 | 0.662 | 0.664 | 0.662 | 0.692 | 0.887 |
| 16 | 0.842 | 0.857 | 0.905 | 0.420 | 0.728 | 0.889 | 0.930 | 0.453 | 0.646 | 0.759 | 0.926 | 0.731 | 0.790 | 0.812 | 0.898 | 0.675 |
| 14 | 0.971 | 0.887 | 0.979 | 0.916 | 0.818 | 0.905 | 0.941 | 0.835 | 0.918 | 0.778 | 0.848 | 0.930 | 0.984 | 0.742 | 0.972 | 0.926 |
| 12 | 0.972 | 0.927 | 0.983 | 0.944 | 0.618 | 0.976 | 0.967 | 0.771 | 0.950 | 0.953 | 0.986 | 0.974 | 0.841 | 0.973 | 0.986 | 0.972 |
| 10 | 0.713 | 0.795 | 0.861 | 0.986 | 0.972 | 0.991 | 0.982 | 0.777 | 0.726 | 0.936 | 0.916 | 0.916 | 0.979 | 0.629 | 0.959 | 0.947 |
| 8 | 0.921 | 0.838 | 0.948 | 0.916 | 0.925 | 0.929 | 0.942 | 0.441 | 0.916 | 0.850 | 0.972 | 0.894 | 0.858 | 0.725 | 0.911 | 0.966 |
| 6 | 0.666 | 0.839 | 0.909 | 0.927 | 0.960 | 0.983 | 0.961 | 0.907 | 0.687 | 0.949 | 0.946 | 0.943 | 0.457 | 0.847 | 0.983 | 0.972 |
| 4 | 0.486 | 0.830 | 0.977 | 0.919 | 0.906 | 0.992 | 0.966 | 0.921 | 0.908 | 0.971 | 0.917 | 0.812 | 0.690 | 0.872 | 0.898 | 0.902 |
| 2 | 0.716 | 0.929 | 0.937 | 0.962 | 0.955 | 0.974 | 0.988 | 0.878 | 0.339 | 0.953 | 0.476 | 0.979 | 0.978 | 0.961 | 0.780 | 0.961 |

Please describe the image in detail.

Output answer: The image features a wooden dining table filled with a variety of foods, …

Figure 3: Uncertainty of different early layers to predict the next token. Rows denote indices of the early layers, and column names are decoded tokens in each step. Uncertainty distribution is dynamic.

modality-shared feature space via an MLP or Q-Former module (Wadekar et al., 2024). Aligned vision tokens are represented as $X_v = \{x_1, x_2, \ldots, x_{n_v}\}$ and used as part of the LLM input alongside the text tokens $X_q = \{x_{n_v+1}, x_{n_v+2}, \ldots, x_{n_v+n_q-1}\}$ that are embedded from tokenized text input by the embedding layer. Subsequently, the vision and text tokens are concatenated as the final input sequence and we denote it as $\{x_i\}_1^{t_n-1}$ where $t_n = n_v + n_q$, which is then fed into successive transformer layers. We denote the output of the $l$-th layer as $h_t^{(l)}$. Then, the vocabulary head $\zeta(\cdot)$ is used to predict the probability of the next token $x_t$ among the vocabulary set $\mathcal{X}$ as follows,

$$p(x_t \mid x_{<t}) = \text{softmax}(\zeta(h_t^{(L)}))_{x_t}, x_t \in \mathcal{X}. \tag{1}$$

## 3.2 WHAT CAUSES HALLUCINATIONS

**A hypothesis on the cause of hallucinations.** Informed by the phenomenon of catastrophic forgetting (Zhai et al., 2023) in MLLMs, we argue that the capabilities of LLMs to comprehend and memorize different modalities are quite distinct. Taking image and text inputs as an example, since an image possesses a much higher information density than a piece of text, it is reasonable to assume that *LLMs struggle to understand and memorize vision tokens compared to text tokens*, prone to fantasies.

**Uncertainty quantification.** Following the DoLa (Chuang et al., 2023), we compute the probability of the next token via the vocabulary head $\zeta$ on each layer during reasoning. Then, we introduce an entropy-based metric (Farquhar et al., 2024) to quantify the output uncertainty as $u = \sum -p_k \log p_k / \log K$, where $\{p_k\}_{i=1}^K$. With uncertainty, we make an important assumption.

**Assumption 3.1.** LUFH: The Lower the Uncertainty, the Fewer the Hallucinations to be generated.

We define $\gamma$ as the threshold that separates high uncertainty from low uncertainty in predictions. The candidate layer with uncertainty exceeding $\gamma$ is termed a *premature layer*. The proof of Assumption 3.1 (LUFH) is provided in Section 4. Based on the hypothesis, we aim to complement visual evidence in MLLMs to eliminate the hallucination caused by visual forgetting. In the following, we discuss our motivation, how MEMVR is implemented, and why it can work in Section 3.3 and 3.4.

## 3.3 RELATIONSHIP BETWEEN HALLUCINATIONS AND UNCERTAINTY

As findings of Chen et al. (2024) in LLMs: "*incorrect tokens generally exhibit higher entropy than correct ones*", we also observe this phenomenon in MLLMs (visualization cases are shown in Appendix C.3). This implies the effectiveness of entropy-based metrics for detecting hallucinations. In this work, we use uncertainty as the metric. We present our in-depth findings in this section.

**Finding #1: In the context of tokens involving *objects, attributes or relations*, uncertainty is high.** We conduct preliminary analysis with 32-layer LLaVA-1.5-7B. Specifically, we compute the uncertainty in the output distributions $p^{(l)}(\cdot \mid x_{<t})$ of early exiting layers. Figure 3 shows the uncertainty scores of different early layers when decoding the answer, we can observe that the computed uncertainty remains relatively high in later layers when predicting key entity objects, attributes, or relations, such as *wooden, filled, and food* in Figure 3. This phenomenon suggests that LLM is still uncertain about its predictions in the last few layers and may inject more factual knowledge into the predictions. On the other hand, when predicting function words and those tokens copied from the question, *e.g.*, *image, a, with*, we observe that the uncertainty becomes very low from the middle layers. This finding implies that the model is deterministic for easy-to-predict tokens at the intermediate layer and keeps the distribution of outputs almost constant at higher layers, however, it is more uncertain for difficult-to-predict key tokens and may constantly change its predictions until the final layer.

Figure 4: The illustration of how dynamic premature layer injection works.

**Finding #2: The uncertainty distribution of different tokens is dynamic during the generation.** As can be seen in Figure 3, when visual factual knowledge is required for the prediction of the next token, such as *wodden, filled, food*, most layers of LLaVA are of higher uncertainty, and appear to shift the prediction of the later layers. The prediction of the next tokens such as quantity, target category, color, relation, etc. requires evident visual knowledge, but not for function words or commas

From a qualitative perspective, intermediate values of uncertainty reflect the thinking or revision process of MLLMs, while too high uncertainty implies that the model is confused and relies on random guesses. For complex input images, the uncertainty of the output distributions tends to rise, which can propagate through subsequent layers, resulting in incorrect predictions, or gibberish.

## 3.4 VISUAL RETRACING AND ITS DYNAMICS

Motivated by the above findings, we propose re-injecting visual evidence during elevated uncertainty in the model's reasoning. This strategy treats visual tokens as anchors to recalibrate off-target predictions and reduces uncertainties in *object, attribute, relationship* tokens. Experimental results also demonstrate that our method reduces uncertainty and alleviates hallucinations as shown in Figure 9. We term this schema of re-injecting visual evidence as "visual retracing" that is to be elaborated further in Section 3.4.1. Further, we design a dynamic injection strategy, detailed in Section 3.4.2, ensuring timely visual evidence when generating uncertain visual-reliant tokens (Figure 4).

### 3.4.1 FFN WITH VISUAL RETRACING

We introduce the implementation of our proposed memory-space visual retracing method. Previous study (Geva et al., 2021) has found that FFN acts as a key-value memory storing factual knowledge. Inspired by the fact that FFN executes analogous retrieval from its key-value memory, we consider "visual retracing" to serve as a simplified and efficient information re-retrieval process (Jie et al., 2024). Concretely, given a hidden token $x \in \mathbb{R}^d$ and dimension-aligned vision tokens $z_v = (z_{v,1}, \ldots, z_{v,n_v})^\top \in \mathbb{R}^{n_v \times d}$, FFN with visual retracing at $l$-th layer can be written as follows,

$$\mathrm{FFN}^{(l)}(x \propto z_v) = (1-\alpha)\,\mathrm{FFN}^{(l)}(x) + \alpha\,\mathrm{Retrace}^{(l)}(z_v \mid x), \tag{2}$$

where $\alpha$ denotes the injection ratio of visual memory (proportional to image complexity), $x \propto z_v$ denotes execute visual retracing from $x$ to visual features $z_v$. Vanilla FFN comprises two FC layers with non-linear activation in between and can be formulated as $\mathrm{FFN}(x) = \phi(xW_1)\,W_2^\top$, $\phi$ is activation function like ReLU or SiLU (Liu et al., 2020). Separately, the weight matrices can be rewritten as: $W_1 = (k_1, k_2, \ldots, k_m) \in \mathbb{R}^{d \times m}, W_2 = (v_1, v_2, \ldots, v_m) \in \mathbb{R}^{d \times m}$, in which $k_i \in \mathbb{R}^d$ and $v_i \in \mathbb{R}^d$ are entries of key and value, respectively. Thus, FFN can be interpreted as using input $x$ as the query to calculate its similarity with keys to search for matching values. Analogously, we consider a simple and efficient retrieval process for visual retracing on $l$-th premature layer as,

$$\mathrm{Retrace}^{(l)}(z_v \mid x) = \sum_{i=1}^{n_v} \phi(\langle x, z_{v,i}\rangle) \cdot z_{v,i}. \tag{3}$$

From the perspective of FFN, visual retracing works by treating $x$ as a query, and $(z_{v,i}, z_{v,i})$ as new key-value entries (visual evidence) to supplement vision-related information in hidden states. In this information re-retrieval process, MemVR does not introduce any parameters that need to be trained.

### 3.4.2 DYNAMIC PREMATURE LAYER INJECTION

To magnify the effectiveness of visual retracing, the optimal premature layer should ideally be the layer most uncertain about probable answers to visual questions. In practice, we consider that the

uncertainty of a candidate layer exceeding the threshold $\gamma$ warrants visual retracing. Inspired by the fact that *early exit* patterns (Teerapittayanon et al., 2016; Elbayad et al., 2020; Schuster et al., 2022) have proven effective in directly employing the language heads $\zeta$ to the hidden states of the middle layers, even without a special training process (Kao et al., 2020), we compute the uncertainty of the next token probability on the early layers for reasoning. As our Finding #2, we utilize layer-specific uncertainty to allow for dynamic premature layer injection at each time step as illustrated in Figure 3.

**Dynamic Layer Injection.** For MLLMs with different numbers of layers, we first partition the layers into several (typically two) buckets according to the total layers for finding a sensible candidate layer set, as detailed in Appendix C.1. Then, as algorithm 1 shown, this dynamic injection strategy identifies desirable premature layers among the candidate layers for visual retracing based on output uncertainty of different layers, thus better leveraging insights from different layers.

**Static Fixed Layer Injection.** In addition to the dynamic premature layer injection strategy, another more straightforward strategy worth considering is to perform a brute-force experiment on all possible early layers using a validation set and selecting the layer with the best average performance. We refer to this simple

---

**Algorithm 1** Dynamic Injection Strategy

1: $h_t, z_v$ denote hidden states and visual evidence. We set trigger = True.
2: **for** $l = 1$ to $L - 1$ **do**
3: $\quad p^{(l)} = \mathrm{softmax}(\zeta(h_t^{(l)}))_{x_t}$
4: $\quad u^{(l)} = \sum -p^{(l)} \log p^{(l)} / \log K$.
5: $\quad$ **if** trigger==True and $u^{(l)} > \gamma$ **then**
6: $\quad\quad$ Execute Retrace$^{(l+1)}(z_v \mid h_t^{(l+1)})$
7: $\quad\quad$ Select FFN$^{(l+1)}(h_t^{(l+1)} \propto z_v)$
8: $\quad\quad$ trigger=False # only once
9: $\quad$ **else**
10: $\quad\quad$ Select FFN$^{(l+1)}(h_t^{(l+1)})$
11: $\quad$ **end if**
12: **end for**

---

strategy as MEMVR-static. However, the MEMVR-static approach presents two limitations: (I) it requires more extensive hyperparameter tuning across layers, and (II) the optimal layer is highly sensitive to data distribution, necessitating an in-distribution validation set. In contrast, our proposed dynamic layer injection strategy mitigates these challenges by reducing the layer search space and improving robustness without depending on in-distribution validation. Empirical comparisons between our MEMVR using dynamic and static strategies are provided in Section 4.2 and Table 4.

3.5 THEORETICALLY UNDERSTANDING WHY MEMVR WORKS

In order to gain further insight into the reasons behind the effectiveness of MEMVR in mitigating hallucinations and its robust performance on general benchmarks, we attempt to explain these phenomena via the following three theorems from an information-theoretic perspective.

**Theorem 3.1.** *Let $H_{vq}$ be the hidden states of FFN and $\hat{H}_{vq}$ be after reinjection of visual evidence $Z_v$. MEMVR enhances Mutual Information (MI) between $\hat{H}_{vq}$ and $Z_v$: $I(\hat{H}_{vq}; Z_v) \geq I(H_{vq}; Z_v)$.*

The reinjection of $Z_v$ at intermediate layers of the model facilitates the replenishment of critical visual information, which may have been lost or distorted through earlier layers. This process increases MI between the hidden states and the visual features, ensuring adequate visual context is preserved.

**Theorem 3.2.** *Let $Y$ be the target output dependent on hidden states. If MI between $H_{vq}$ and $Z_v$ increases, then conditional entropy $H(Y \mid H_{vq}^{(l)})$ decreases with $H(Y \mid \hat{H}_{vq}) \leq H(Y \mid H_{vq})$.*

According to Theorem 3.1, 3.2, and DPI (Cover et al., 1991), MEMVR improves the quality of the representations $H_{vq}$ and reduces the uncertainty in the output $Y$. As a result, the probability of hallucinations decreases, which is consistent with the observed findings of hallucination mitigation.

**Theorem 3.3.** *Within the Information Bottleneck (IB) framework, the loss of objective function, represented by the notation $\mathcal{L}(T)$, is optimized by MEMVR, which is defined as $\mathcal{L}(\hat{H}_{vq}) \leq \mathcal{L}(H_{vq})$, where $\mathcal{L}(H_{vq}) = I(H_{vq}; X_{vq}) - \beta I(H_{vq}; Y)$ is IB loss, and $beta$ is a trade-off parameter.*

Anchored in the information bottleneck framework, MEMVR optimizes the delicate balance between retaining relevant information from multimodal inputs and compressing non-essential details, thereby safeguarding the predictive performance of the hidden representations for the target output $Y$.

The theoretical underpinnings of MEMVR are supported by the Data Processing Inequality (Cover et al., 1991) and the contraction properties of stochastic mappings in deep neural networks, as shown in various studies on the Information Bottleneck Principle (Achille & Soatto, 2018). By enhancing mutual information and reducing the uncertainty in hidden states, MEMVR effectively mitigates hallucinations while preserving computational efficiency. Detailed proofs are in Appendix B.

| Method | MSCOCO | | A-OKVQA | | GQA | |
|---|---|---|---|---|---|---|
| | %Accuracy | %F1 Score | %Accuracy | %F1 Score | %Accuracy | %F1 Score |
| LLaVA1.5-7B | 81.38 ↑0.0 | 79.65 ↑0.0 | 79.13 ↑0.0 | 79.10 ↑0.0 | 79.00 ↑0.0 | 79.13 ↑0.0 |
| + VCD (Leng et al., 2024) | 84.66 ↑3.3 | 84.52 ↑4.9 | 80.99 ↑1.8 | 82.30 ↑3.2 | 81.74 ↑2.7 | 82.16 ↑3.0 |
| + OPERA (Huang et al., 2024a) | 84.77 ↑3.4 | 85.46 ↑5.8 | 84.27 ↑5.1 | 84.08 ↑5.0 | 84.03 ↑5.0 | 83.83 ↑4.7 |
| **+ MEMVR (Ours)** | **87.00 ↑5.7** | **85.87 ↑6.2** | **86.21 ↑7.0** | **86.64 ↑7.5** | **85.25 ↑6.2** | **85.59 ↑6.4** |
| Qwen-VL-10B | 83.79 ↑0.0 | 81.13 ↑0.0 | 84.74 ↑0.0 | 83.27 ↑0.0 | 84.41 ↑0.0 | 82.66 ↑0.0 |
| + VCD (Leng et al., 2024) | 84.27 ↑0.4 | 82.12 ↑1.0 | 84.09 ↓0.7 | 82.53 ↓0.7 | 83.73 ↓0.7 | 82.75 ↑0.1 |
| + OPERA (Huang et al., 2024a) | **84.93 ↑1.1** | **83.41 ↑2.3** | - | - | - | - |
| **+ MEMVR (Ours)** | 84.07 ↑0.3 | 81.55 ↑0.4 | **86.43 ↑1.8** | **85.56 ↑2.3** | **85.69 ↑1.3** | **84.53 ↑1.9** |

Table 2: Performance evaluation on POPE. The best results in each scenario are **bolded** for clarity. We report the averages under the three settings, e.g., *Random*, *Popular*, and *Adversarial* to show the robustness of the different methods directly. Green denotes improvement, and Red means degradation.

| Method | Commonsense QA(Reasoning) | Object-level Hallucination | | Attribute-level Hallucination | | Total Scores |
|---|---|---|---|---|---|---|
| | | Existence | Count | Position | Color | |
| LLaVA1.5-7B | 110.71 ↑0.0 | 175.67 ↑0.0 | 124.67 ↑0.0 | 114.00 ↑0.0 | 151.00 ↑0.0 | 676.05 |
| + VCD (Leng et al., 2024) | 112.86 ↑9.9 | 184.66 ↑9.0 | 138.33 ↑13.6 | 128.67 ↑14.6 | 153.00 ↑2.0 | 717.52 |
| + OPERA (Huang et al., 2024a) | 115.71 ↑5.5 | 180.67 ↑5.0 | 133.33 ↑8.6 | 123.33 ↑9.3 | 155.00 ↑4.0 | 708.04 |
| **+ MEMVR (Ours)** | **121.42 ↑18.5** | **190.00 ↑14.3** | **155.00 ↑30.3** | **133.33 ↑19.3** | **170.00 ↑19.0** | **769.75** |
| Qwen-VL-10B | 106.40 ↑0.0 | 155.00 ↑0.0 | 127.67 ↑0.0 | 131.67 ↑0.0 | 173.00 ↑0.0 | 693.74 |
| + VCD (Leng et al., 2024) | 104.33 ↓2.1 | 156.00 ↑1.0 | 131.00 ↑3.3 | 128.00 ↓3.6 | 181.67 ↑8.6 | 701.00 |
| + OPERA (Huang et al., 2024a) | 104.33 ↑2.2 | 165.00 ↑6.9 | 145.00 ↑4.8 | 133.33 ↑1.6 | 180.00 ↑7.0 | 727.66 |
| **+ MEMVR (Ours)** | **120.00 ↑13.6** | **185.00 ↑30.0** | **145.00 ↑17.3** | 123.33 ↓8.3 | **185.00 ↑12.0** | **758.33** |

Table 3: Results on the hallucination subset of MME (including commonsense reasoning, existence, count, position, color scores). The best are in **bold**. MemVR achieves dramatic improvements.

## 4 EXPERIMENTS

This section details the evaluation of our MEMVR across three MLLMs on seven benchmarks.

### 4.1 EXPERIMENT SETUP

**Datasets and Metrics.** To rigorously assess the effectiveness of our proposed method, we conduct a comprehensive set of experiments across two benchmarks specifically designed to evaluate hallucination mitigation and five general-purpose benchmarks to gauge the general performance:

❶ Hallucination benchmarks: Polling-based Object Probing Evaluation (POPE) (Li et al., 2023b), and Caption Hallucination Assessment with Image Relevance (CHAIR) (Rohrbach et al., 2018);

❷ General-purpose benchmarks: VizWiz-VQA (Gurari et al., 2018), MLLM Comprehensive Evaluation (MME) (Fu et al., 2023), Multimodal Benchmark (MMBench) (Liu et al., 2023b), Multimodal Veterinarian (MM-Vet) (Yu et al., 2024c), LLaVA-Bench (in-the-wild) (Liu et al., 2024).

More detailed information on these various benchmarks can be obtained from the Appendix C.1.

**Backbones and Baselines.** To evaluate our method, we utilize three well-known MLLMs: LLaVA-1.5 (Liu et al., 2024), Qwen-VL (Bai et al., 2023), and GLM4V (Wang et al., 2023). Further, We compare our methods with classic training-tree SOTA methods designed to mitigate object hallucination, including visual contrastive decoding SOTA VCD Leng et al. (2024), OPERA (Huang et al., 2024a) based on overconfidence penalty and hindsight allocation. As Dola (Chuang et al., 2023) is layer-wise contrastive decoding for LLMs and performs poorly in MLLMs, it will not be shown in the experiment. Experimental results are obtained and benchmarked using unified implementation.

**Implementation Details.** Greedy search is used as the default decoding strategy in MEMVR for all benchmarks. For benchmarks, annotation questions are adapted to MLLM templates. For POPE, COCO, A-OKVQA, and GQA are used, while MMBench_DEV_EN is used for MMBench. MM-Vet is assessed using MM-Vet Online Evaluator, and gpt4-1106-preview is used for LLaVA-Bench. CHAIR uses images from COCO Val2014 with the query "Please describe this image in detail". In MEMVR, do_sample=False, temperature=0, threshold=0.75, beam=1. All settings of the compared method follow the default configurations from the original papers. More details are in Appendix C.1.

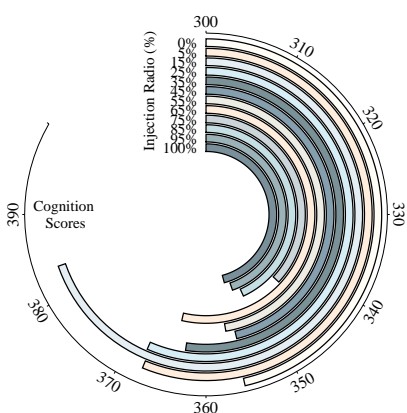

Figure 5: Results on MMBench. MEMVR enhances comprehensive performance on diverse tasks.

| Strategy | Static-3 | Static-7 | Static-11 | Static-15 | Static-19 | Static-23 | Static-27 | Static-31 | **Dynamic** |
|---|---|---|---|---|---|---|---|---|---|
| Perception | 1508.08 | 1500.48 | 1526.40 | **1529.07** | 1513.96 | 1500.29 | 1510.47 | 1510.34 | 1512.80 |
| Cognition | 364.64 | 347.14 | 362.86 | 352.14 | 350.36 | 357.86 | 355.71 | 346.43 | **383.92** |
| Total Score | 1872.72 | 1847.62 | 1889.26 | 1881.22 | 1864.32 | 1858.15 | 1866.18 | 1856.77 | **1896.72** |

Table 4: Results of MEMVR-static and MEMVR-dynamic on MME. Static-# indicates fixation on layer # for visual retracing. MEMVR-dynamic achieves optimal performance improvements.

## 4.2 QUANTITATIVE RESULTS

**Key Finding: MEMVR consistently outperforms the baselines in mitigating hallucinations and improving overall accuracy across various scenarios.**

**MEMVR performance on hallucination benchmarks.** We conduct POPE, CHAIR, and MME evaluations, as shown in Table 2, Table 3 and Table 5, our MEMVR obviously surpasses all of the compared baselines. For the results of POPE evaluation, we observe that our proposed MEMVR presents robust effects. The performance of MEMVR surpasses the baseline results by large margins, *i.e.*, average up to +7.0% accuracy and +7.5% F1 score on A-OKVQA dataset under *Random*, *Popular*, and *Adversarial* settings. As showcased in Table 5, our MEMVR achieves up to 15.6% improvements on CHAIR$_I$ compared with vanilla LLaVA-1.5-7B. For MME subset evaluations (encompass both object-level and attribute-level hallucinations), results in Table 3 show that MEMVR achieves a uniform improvement in handling object-level and attribute-level hallucinations, as well as commonsense reasoning. *Existence*, *Count*, and *Color* scores all achieve dramatic improvements (*Existence* score in Qwen-VL up 30). On the contrary, *Position* scores are relatively low, which suggests weak position reasoning capability in MLLMs.

Figure 6: Comparison of different injection ratios $\alpha$ on cognition scores.

**MEMVR performance on general-purpose benchmarks.** We evaluate the performance of MEMVR on general-purpose benchmarks, *i.e.*, VizWiz, MME, MMBench, MM-Vet, and LLaVA-Bench. Appx. C summarizes the results on MMBench, highlighting MEMVR's comparative performance relative to SOTA methods. As shown in Table 4.2, MEMVR

| Model | Length | CHAIR$_S$ ↓ | CHAIR$_I$ ↓ | Recall ↑ |
|---|---|---|---|---|
| LLaVA-1.5 | 100.6 | 50.0 ↑0.0 | 15.4 ↑0.0 | 77.1 ↑0.0 |
| + VCD | 100.4 | 48.6 ↓1.4 | 14.9 ↓0.5 | 77.3 ↑0.2 |
| + OPERA | 98.6 | 47.8 ↓2.2 | 14.6 ↓0.8 | 76.8 ↓0.3 |
| + MEMVR | 99.6 | **46.6 ↓3.4** | **13.0 ↓2.4** | **80.8 ↑3.7** |

Table 5: CHAIR hallucination evaluation results of LLaVA. Small values correspond to fewer hallucinations.

consistently outperforms competing models. Besides, MEMVR achieves a significant improvement in overall performance listed in Table 7, with an average increase of 6.1% in OCR and spatial awareness tasks, demonstrating superior generalization capabilities. These results indicate that compared with CD-based methods, MEMVR excels in hallucination mitigation and delivers competitive performance on general-purpose benchmarks. More complete results are in Appendix C.

**The efficiency of MEMVR.** MEMVR operates dynamically based on the uncertainty, which employs visual retracing when the uncertainty exceeds threshold $\gamma$ on the early layer. If the uncertainty remains low across all layers—indicating that the model is highly confident in its generated results—MEMVR is not triggered. This mechanism ensures efficient inference without extra com-

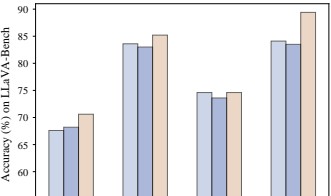 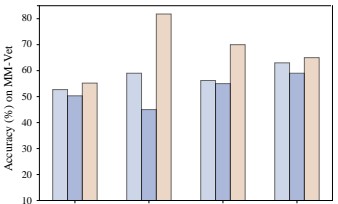 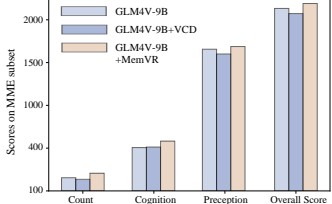

Figure 8: Results of GLM4V-9B. MEMVR enhances comprehensive performance on diverse tasks.

| Method | gn_kw_rec | rec | ocr_sp | ocr | ocr_sp_rec | ocr_kw_rec | ocr_gn_sp | Total |
|---|---|---|---|---|---|---|---|---|
| LLaVA1.5-7B | 18.1 ↑0.0 | 67.6 ↑0.0 | 17.7 ↑0.0 | 48.3 ↑0.0 | 60.0 ↑0.0 | 21.2 ↑0.0 | 10.0 ↑0.0 | 31.1 ↑0.0 |
| + VCD | 19.2 ↑1.1 | 62.2 ↓5.4 | 15.8 ↓1.9 | 29.2 ↓20.0 | 42.5 ↓17.5 | 17.5 ↓3.7 | 60.0 ↑50.0 | 30.2 ↓1.1 |
| + OPERA | 21.8 ↑3.7 | 61.9 ↓5.7 | 21.5 ↑3.8 | 51.7 ↑3.4 | 56.2 ↓3.8 | 11.2 ↓10.0 | 30.0 ↑20.0 | 32.0 ↑0.9 |
| + MemVR | 19.5 ↑1.4 | 70.3 ↑2.7 | 23.8 ↑6.1 | 48.3 ↑0.0 | 58.8 ↓1.2 | 21.2 ↑0.0 | 30.0 ↑20.0 | 32.4 ↑1.3 |

Table 7: MM-Vet evaluation results with multiple complicated multimodal tasks, where *gn* denotes language generation, *kw* means knowledge, *sp* denotes spatial awareness, and *rec* is recognition.

putational overhead. Compared with VCD and OPERA, they need inference twice or the rollback strategy leads to exponentially added overheads, our MEMVR only once regular inference.

**MEMVR performance under different qualities of visual features.** To examine the impact of visual features on MEMVR 's performance, we introduced Gaussian noise (McHutchon & Rasmussen, 2011) into the extracted visual features when visual retracing. We gradually increased the noise level to assess how MEMVR 's performance would respond, using MME as the benchmark in LLaVA1.5-7B. As illustrated in Figure 7, both the perception and cognition scores declined as the noise step increased. At high noise, the performance fell significantly. This demonstrates that MEMVR is sensitive to the quality of visual features, and can efficiently understand shallow visual features.

**MEMVR performance under different injection ratios** α**.** MEMVR leverages layer entropy to trigger visual retracing dynamically. We compared the performance of fixed retracing layers with dynamic retracing layers and found that dynamic retracing outperformed fixed retracing, as demonstrated in Table 4. Under all experimental conditions, dynamic retracing achieved the highest total score on the MME evaluation using LLaVA. Furthermore, we analyzed

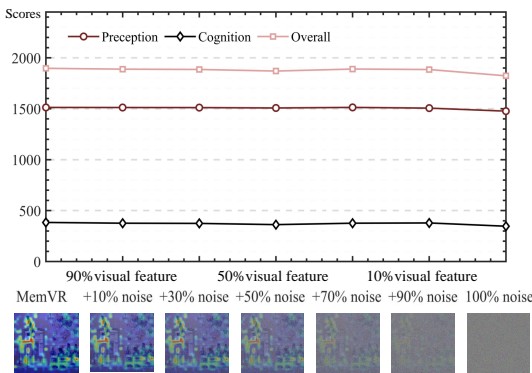

Figure 7: MME evaluation results under different mixing ratios of noise and visual features.

| Model | Convs ↑ | Detail ↑ | Complex ↑ | All ↑ |
|---|---|---|---|---|
| LLaVA-1.5 | 58.8 ↑0.0 | 52.1 ↑0.0 | 74.6 ↑0.0 | 63.4 ↑0.0 |
| + VCD | 57.8 ↓1.0 | 50.8 ↓1.3 | 77.9 ↑3.3 | 59.1 ↓4.3 |
| + OPERA | 59.5 ↑0.7 | 49.6 ↓2.5 | 78.6 ↑4.0 | 59.8 ↓3.6 |
| + MEMVR | 63.8 ↑5.0 | 52.6 ↑0.5 | 77.9 ↑3.3 | 64.0 ↑0.6 |

Table 6: LLaVA-Bench evaluation results.

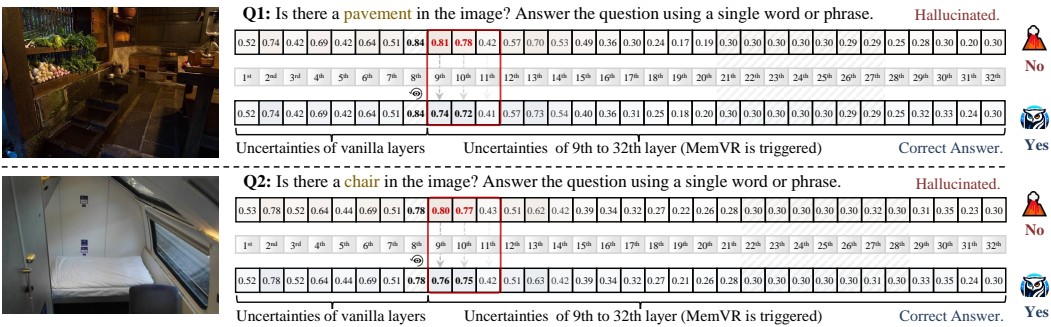

Figure 9: Visualisation of uncertainty across layers without and with MEMVR. MEMVR effectively reduces uncertainty after 8th layer, contributing to hallucination mitigations as Assumption 3.1.

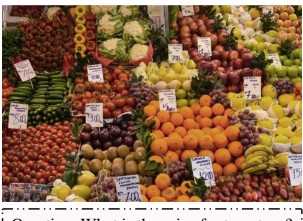

**Ground Truth Caption:** 8.0

**GLM-4v-9B Default:** The price for tomatoes is 9.00, as indicated by the white sign with black lettering among the red tomatoes.

**GLM-4v-9B + VCD:** The price for tomatoes is not clearly visible in the image provided. There are price tags for some of the other fruits and vegetables, but the tomatoes are either not labeled or the label is not in focus. If you need the exact price for tomatoes, I would recommend visiting the market in person or checking with the vendor.

**GLM-4v-9B + MemVR (Ours):** The price for tomatoes is 8.00, as indicated by the white sign with black lettering among the red tomatoes.

Question: What is the price for tomatoes?

Figure 10: A case study in long text generation. MEMVR effectively mitigates hallucinations.

the injection ratio for MEMVR, with the optimal injection ratio observed around 0.15, as shown in Table 4. This indicates that a balanced level of $\alpha$ is crucial for maximizing performance.

### 4.3 QUALITATIVE STUDY: WHAT TYPES OF FAULTS CAN OUR METHOD ADDRESS?

In addition to evaluating single-word question-answering (QA) benchmarks, we further explore the models' ability to generate comprehensive long-text descriptions for various tasks. As depicted in Figure 10, our MEMVR excels in accurately identifying question-relevant details within images. In contrast, as discussed in Appendix C, Qwen-VL-Chat exhibits occasional difficulties in producing detailed image descriptions when applied to VCD. These shortcomings become evident in scenarios requiring a nuanced interpretation of image content. This suggests that MEMVR demonstrates superior adaptability across different architectures, enabling more reliable long-text generation.

### 4.4 LIMITATIONS AND FURTHER DISCUSSIONS

While MEMVR demonstrates significant promise in improving the performance in MLLMs, it is not without limitations. One major challenge lies in the variability of MLLM architectures. Different MLLMs employ varying FNNs, activation functions, and knowledge systems, making it difficult to identify the optimal hyperparameters—such as injection ratios $\alpha$, and *premature* layer for each specific model. This requires considerable effort in model-specific tuning, which may limit the scalability of MEMVR without further automation or standardization in hyperparameter selection.

Can MEMVR adapt to other MLLMs? Definitely, MEMVR is designed to be flexible and compatible with various architectures. A key advantage of MEMVR lies in its ability to function without requiring modifications to the Transformers library, facilitating smooth integration into both older and cutting-edge models. We conducted extensive experiments across multiple benchmarks with LLaVA, Qwen-VL, and GLM-4V, and all three models outperformed baselines.

Additionally, although our research focused on MLLMs with image inputs, MEMVR is theoretically applicable to LLMs with different modalities. However, we have yet to explore its performance on inputs such as voice, point clouds, 3D meshes, or video frames. This opens up an exciting avenue for future work, where we plan to extend MEMVR 's framework to these diverse input formats and assess its efficacy across a broader range of tasks. Furthermore, we will explore integrating MEMVR into the training procedures of MLLMs, rather than limiting its application to inference, to evaluate whether this could lead to even greater improvements in model performance and generalization.

## 5 CONCLUSION

This paper proposes a novel training-free paradigm to mitigate hallucination, named MEMVR. In contrast to previous CD-based methods, which primarily focus on eliminating biases of language priors, MEMVR seeks to replenish question-relevant visual clues towards more evidential answers, which signifies the other side of the coin. Our experiments, conducted on seven benchmarks, demonstrate the effectiveness of MEMVR in mitigating hallucination and improving general performance.

**Reproducibility.** To promote transparency and ensure the reproducibility of our work, we release all experimental code, datasets, and detailed tutorials necessary for replicating our experiments in the Supplementary Material. Our goal is to make it straightforward for researchers and practitioners to reproduce our results, regardless of their technical background. Additionally, by providing comprehensive documentation and clear guidelines, we aim to facilitate the extension of our method to other models and architectures, enabling the broader research community to explore its potential applications and improvements. **Ethics Statement.** We illustrate this in the Appendix A.

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

The organization of the appendix is as follows:

## A  ETHIC CONSIDERATIONS

We list some key ethical considerations of our method:

*Bias and fairness*. By injecting visual features from CLIP into the Feed-Forward Network (FFN) layers of LLMs, there's a potential for inherited biases from the original models. CLIP, like many pre-trained models, may contain biases in how it represents certain objects, scenes, or demographics. These biases could propagate, affecting performance on different types of data (e.g., gender, ethnicity, cultural contexts). It's important to evaluate how MemVR performs across diverse datasets and ensure that it doesn't reinforce harmful stereotypes or disproportionately fail for certain groups.

*Misuse of Enhanced Models*. As MemVR aims to improve the long-text generation and overall performance of VLMs, enhanced models could be misused to generate deceptive or misleading content, such as deepfakes or disinformation. It's important to consider whether there are safeguards in place to prevent malicious use of these improved models in scenarios like automated misinformation campaigns or unethical surveillance.

*Data Privacy*. If the benchmarks used for evaluating MemVR include datasets with personally identifiable information or sensitive content, care should be taken to ensure data privacy. Models should be evaluated on publicly available, anonymized, or ethically sourced datasets to avoid violating privacy laws or ethical norms.

## B  THEORETICAL ANALYSIS OF MEMVR IN MLLMS

In Multimodal Large Language Models (MLLMs), hallucinations often arise due to insufficient alignment between visual inputs and the model's internal representations. This paper provides a rigorous theoretical analysis demonstrating that re-injecting visual features into the intermediate layers of MLLMs mitigates hallucinations and enhances representation capability.

We demonstrate that MemVR increases Mutual Information (MI) between the hidden states and visual tokens, decreasing the conditional entropy of outputs given the hidden state for fidelity to the visual input. We begin by defining the relevant variables and information-theoretic concepts that will be used throughout the proof as, $X_{vq}$ denote concatenated tokens of text and vision, with probability distribution $p(X_{vq})$; $Z_v$ means visual (image) features, with probability distribution $p(Z_v)$; The output hidden states of the Transformer model at layer $k$, defined recursively as: $H_{vq}^{(k)} = f^{(k)}(H_{vq}^{(k-1)}, \mathbf{1}_{k=m} Z_v)$, where $\mathbf{1}_{k=m}$ is the indicator function that equals 1 when $k = m$ (the layer where $Z_v$ is rejected) and 0 otherwise, and $Y$ denotes the target output of MLLMs.

The probability of hallucination can be expressed as:

$$P_{\text{hallucination}} = P(Y \neq Y^* \mid X_{vq}),$$

where $Y^*$ is the ground truth output. According to information theory, a higher conditional entropy $H(Y \mid X_{vq})$ indicates greater uncertainty of $Y$ given $X_{vq}$, which increases the probability of hallucination.

**Information Flow of Visual Features.** In a standard Transformer model, the initial input $X_{vq}$ undergoes multiple layers of processing. As the number of layers increases, the initial visual information may gradually diminish (*information forgetting*). In the absence of MemVR, the MI between the hidden states and the visual features $Z_v$ tends to decrease with depth:

$$I(H_{vq}^{(l)}; Z_v) \leq I(H_{vq}^{(l-1)}; Z_v),$$

for $l > 1$. This inequality indicates that in deeper layers, $H_{vq}^{(l)}$ contains less vision-related information.

**Theorem B.1.** *Assume that each Transformer layer acts as a deterministic or stochastic mapping with the Markov property. Then, the mutual information between the hidden states and the visual features decreases with depth:*

$$I(H_{vq}^{(l)}; Z_v) \leq I(H_{vq}^{(l-1)}; Z_v).$$

*Proof.* Each Transformer layer can be modeled as a stochastic mapping (Markov kernel) that processes the input hidden states. Specifically, $H_{vq}^{(l)}$ is a function of $H_{vq}^{(l-1)}$, possibly incorporating additional inputs such as $Z_v$ at specific layers.

According to the **Data Processing Inequality (DPI)** (Cover et al., 1991), if $A \to B \to C$ forms a Markov chain, then:
$$I(A; C) \leq I(A; B).$$

In this context, consider $A = Z_v$, $B = H_{vq}^{(l-1)}$, and $C = H_{vq}^{(l)}$. Since $H_{vq}^{(l)}$ is generated from $H_{vq}^{(l-1)}$ without direct access to $Z_v$, we have the Markov chain $Z_v \to H_{vq}^{(l-1)} \to H_{vq}^{(l)}$. Applying DPI yields:

$$I(Z_v; H_{vq}^{(l)}) \leq I(Z_v; H_{vq}^{(l-1)}).$$

Thus, mutual information between the hidden states and the visual features does not increase with depth. $\square$

**Visual Retracing in MLLMs.** We reinject vision tokens $Z_v$ on $l$-th layer ($ahead\_layer \leq l < L$):

$$\hat{H}_{vq}^{(l)} = \text{FFN}^{(l)}(H_{vq}^{(l)} \propto Z_v).$$

MemVR ensures that after the $l$-th layer, $\hat{H}_{vq}^{(l)}$ again contains question-aligned visual information.

**Theorem B.2.** *Let $H_{vq}$ be the hidden states of FFN and $\hat{H}_{vq}$ be after reinjection of visual evidence $Z_v$.* MEMVR *enhances Mutual Information (MI) between $\hat{H}_{vq}$ and $Z_v$: $I(\hat{H}_{vq}; Z_v) \geq I(H_{vq}; Z_v)$.*

*Proof.* We aim to show that reinjecting $Z_v$ at layer $l$ increases the mutual information between the hidden states and $Z_v$ conditioned on $X_{vq}$.

By the definition of conditional mutual information:

$$I(\hat{H}_{vq}^{(l)}; Z_v \mid X_{vq}) = \mathbb{E}_{X_{vq}}[I(\hat{H}_{vq}^{(l)}; Z_v \mid X_{vq} = x)].$$

Similarly,
$$I(H_{vq}^{(l)}; Z_v \mid X_{vq}) = \mathbb{E}_{X_{vq}}[I(H_{vq}^{(l)}; Z_v \mid X_{vq} = x)].$$

Given $\hat{H}_{vq}^{(l)} = \text{FFN}_{\diamond}^{(l)}(H_{vq}^{(l)} \propto Z_v)$ denotes the hidden states after utilizing MemVR on $l$-th, reinjection of $Z_v$ introduces a direct dependency between $\hat{H}_{vq}^{(l)}$ and $Z_v$ beyond what is present in $H_{vq}^{(l)}$. Since $\text{FFN}_{\diamond}^{(l)}$ is a deterministic function that incorporates $Z_v$, the mutual information $I(\hat{H}_{vq}^{(l)}; Z_v \mid X_{vq})$ is at least as large as $I(H_{vq}^{(l)}; Z_v \mid X_{vq})$. $\hat{H}_{vq}^{(l)}$ retains all information in $H_{vq}^{(l)}$ and additionally incorporates information from $Z_v$. Thus, MemVR ensures that:

$$I(\hat{H}_{vq}^{(l)}; Z_v \mid X_{vq}) \geq I(H_{vq}^{(l)}; Z_v \mid X_{vq}).$$

By directly incorporating $Z_v$ into the computation of $\hat{H}_{vq}^{(m)}$, MemVR ensures that the hidden states retain more information about the visual features relative to the original hidden states $H_{vq}^{(m)}$, thereby increasing $I(\hat{H}_{vq}^{(m)}; Z_v \mid X_{vq})$, enhancing the representation capability and utilizing visual information. $\square$

**Theorem B.3.** *Let $Y$ be the target output dependent on hidden states. If MI between $H_{vq}^{(l)}$ and $Z_v$ increases, then conditional entropy $H(Y \mid H_{vq}^{(l)})$ decreases, leading to a lower probability of hallucinations:*

$$H(Y \mid \hat{H}_{vq}^{(l)}) \leq H(Y \mid H_{vq}^{(l)}).$$

*Proof.* We aim to show that an increase in mutual information between $\hat{H}_{vq}^{(l)}$ and $Z_v$ conditioned on $X_{vq}$ leads to a decrease in the conditional entropy $H(Y \mid \hat{H}_{vq}^{(l)})$. According to the definition of conditional entropy, we have,

$$H(Y \mid \hat{H}_{vq}^{(l)}) = H(Y) - I(Y; \hat{H}_{vq}^{(l)}),$$

$$H(Y \mid H_{vq}^{(l)}) = H(Y) - I(Y; H_{vq}^{(l)}).$$

From Theorem B.2: $\hat{H}_{vq}^{(l)}$ contains more information about $Z_v$, i.e., $I(\hat{H}_{vq}^{(l)}; Z_v) \geq I(H_{vq}^{(l)}; Z_v)$. There is $I(Y; \hat{H}_{vq}^{(l)}) \propto I(\hat{H}_{vq}^{(l)}; Z_v)$, thus we have $I(\hat{H}_{vq}^{(l)}; Y) \geq I(H_{vq}^{(l)}; Y)$. Then, we assume a dependency between $Z_v$ and $Y$, i.e., $I(Z_v; Y) > 0$, and subtract the inequalities, have:

$$\begin{aligned}
H(Y \mid \hat{H}_{vq}^{(l)}) &= H(Y) - I(Y; \hat{H}_{vq}^{(l)}) \\
&\leq H(Y) - I(Y; H_{vq}^{(l)}) \\
&= H(Y \mid H_{vq}^{(l)}).
\end{aligned}$$

Thus, MemVR reduces the conditional uncertainty of the target output given the intermediate embedding, thereby mitigating the probability of hallucinations and improving the model's predictive capability. $\qquad\square$

**Theorem B.4.** *Within the Information Bottleneck (IB) framework, reinjecting $Z_v$ at layer $m$ optimizes the objective function:*

$$\mathcal{L}(\hat{H}_{vq}^{(m)}) \leq \mathcal{L}(H_{vq}^{(m)}),$$

*where the IB objective is defined as:*

$$\mathcal{L}(H) = I(H; X_{vq}) - \beta I(H; Y),$$

*and $\beta$ is a trade-off parameter.*

*Proof.* The Information Bottleneck (IB) objective aims to find a representation $H$ that maximizes the mutual information with the target $Y$ while minimizing the mutual information with the input $X_{vq}$. The optimization objectives before & after MemVR are as follows:

$$\mathcal{L} = I(H_{vq}^{(l)}; X_{vq}) - \beta I(H_{vq}^{(l)}; Y),$$

$$\mathcal{L}_\diamond = I(\hat{H}_{vq}^{(l)}; X_{vq}, Z_v) - \beta I(\hat{H}_{vq}^{(l)}; Y),$$

where $I(\hat{H}_{vq}^{(l)}; X_{vq}, Z_v) = I(\hat{H}_{vq}^{(l)}; X_{vq}) + I(\hat{H}_{vq}^{(l)}; Z_v \mid X_{vq})$. The gap in the objective function is:

$$\begin{aligned}
\Delta\mathcal{L} &= \mathcal{L}_\diamond^{(l)} - \mathcal{L}^{(l)} \\
&= [I(\hat{H}_{vq}^{(l)}; X_{vq}) + I(\hat{H}_{vq}^{(l)}; Z_v \mid X_{vq}) - \beta I(\hat{H}_{vq}^{(l)}; Y)] - [I(H_{vq}^{(l)}; X_{vq}) - \beta I(H_{vq}^{(l)}; Y)] \\
&= [I(\hat{H}_{vq}^{(l)}; X_{vq}) - I(H_{vq}^{(l)}; X_{vq})] + I(\hat{H}_{vq}^{(l)}; Z_v \mid X_{vq}) - \beta[I(\hat{H}_{vq}^{(l)}; Y) - I(H_{vq}^{(l)}; Y)].
\end{aligned}$$

To ensure that $\mathcal{L}_\diamond^{(m)} \leq \mathcal{L}^{(m)}$, we require: $\Delta\mathcal{L} \leq 0$. We define the changes in mutual information. Let $\Delta I_X = I(H_{vq}^{(l)}; X_{vq}) - I(H_{vq}^{(l-1)}; X_{vq})$, $\Delta I_Y = I(H_{vq}^{(l)}; Y) - I(H_{vq}^{(l-1)}; Y)$. Note that $I(H_{vq}^{(l)}; Z_v \mid X_{vq}) \geq 0$. For $\Delta I_X$, the change in mutual information between $H_{vq}^{(l)}$ and $X_{vq}$ depends on how much additional information from $Z_v$ affects the dependence on $X_{vq}$. We denote the maximum possible increase as $\Delta I_X^{\max}$. For $\Delta I_Y$, From Theorem B.2, $\Delta I_Y \geq 0$, and suppose we can establish a minimum increase $\Delta I_Y^{\min} > 0$. $I(H_{vq}^{(l)}; Z_v \mid X_{vq})$ represents supplement information about $Z_v$ in $H_{vq}^{(l)}$ that is not already explained by $X_{vq}$, and we denote this maximum as $I_{\max}^{Z|X}$.

To satisfy this inequality, choose $\beta$ such that:

$$\Delta\mathcal{L} \leq 0 \Rightarrow \beta \Delta I_Y \geq \Delta I_X + I(H_{vq}^{(l)}; Z_v \mid X_{vq}). \tag{4}$$

Upper Bound on $\Delta I_X$ and $I(H_{vq}^{(l)}; Z_v \mid X_{vq})$ as $\Delta I_X \leq \Delta I_X^{\max}$, $I(H_{vq}^{(l)}; Z_v \mid X_{vq}) \leq I_{\max}^{Z|X}$. Lower Bound on $\Delta I_Y$ as: $\Delta I_Y \geq \Delta I_Y^{\min} > 0$. Then, we derive the condition with error bounds, for $\Delta\mathcal{L} \leq 0$, it suffices that:

$$\beta \Delta I_Y^{\min} \geq \Delta I_X^{\max} + I_{\max}^{Z|X} \Rightarrow \beta \geq \frac{\Delta I_X^{\max} + I_{\max}^{Z|X}}{\Delta I_Y^{\min}}. \tag{5}$$

This condition provides a lower bound for $\beta$ to ensure that reinjecting $Z_v$ at layer $m$ decreases the IB objective function. By adhering to this condition, MemVR optimizes the IB objective, balancing the trade-off between the compression of input information and the preservation of relevant information for prediction. □

By reducing the IB objective function, the model focuses more on information relevant to predicting $Y$ while compressing irrelevant information. The enhanced mutual information with $Y$ reduces the likelihood of generating hallucinated outputs not supported by the visual input.

**Error Bounds Provide Guarantees.** The upper and lower bounds on mutual information changes ensure that, under specific conditions (e.g., the selection of $\beta$), theoretical improvement holds.

**Estimating the Bounds.**

- $\Delta I_Y^{\min}$ requires knowledge of how much additional information about $Y$ is gained by reinjecting $Z_v$. It can be estimated based on the mutual information $I(Z_v; Y)$ and the effectiveness of $H_{vq}^{(m)}$ in capturing information relevant to $Y$.

- $\Delta I_X^{\max}$ can be bounded based on the capacity of $H_{vq}^{(m)}$ to represent $X_{vq}$. Specifically, it relates to how much additional information $H_{vq}^{(m)}$ can encode about $X_{vq}$ beyond what was already captured in $H_{vq}^{(m-1)}$.

- $H(Z_v)$ is bounded by the entropy of the visual features, as mutual information cannot exceed the entropy of $Z_v$.

Through detailed mathematical derivations and the inclusion of upper and lower error bounds, we have established that:

(a) **Increased Mutual Information:** Reinjecting visual features at an intermediate layer increases the mutual information between the model's embeddings and the visual input.

(b) **Reduced Conditional Entropy:** MemVR reduces the conditional uncertainty of the target output given the intermediate embedding, enhancing the model's predictive accuracy and mitigating hallucination phenomena caused by the forgetting of visual information.

(c) **Optimization within IB Framework:** Within the Information Bottleneck framework, MemVR optimizes the objective function, provided certain conditions on the mutual information changes are met and appropriate choices of the trade-off parameter $\beta$ are made.

These theoretical findings provide strong support for the practice of MemVR in MLLMs to improve their performance and reliability.

## C ADDITIONAL EXPERIMENTS, RESULTS, AND DISCUSSIONS

### C.1 BENCHMARKS AND CANDIDATE LAYERS

In this appendix, we provide additional details into the benchmarks referenced in the main paper. To evaluate hallucinations, we employ the following five benchmarks:

**CHAIR** Rohrbach et al. (2018) evaluates how well the generated captions align with the content of the given image. CHAIR consists of two versions: CHAIR_S, which measures the inaccuracies at the sentence level, and CHAIR_I, which evaluates at the object level within the sentence by comparing the number of false objects to the total number of objects. For evaluation, we use the val2014 split of the MSCOCO Lin et al. (2014) dataset, which includes annotations for 80 object categories. We randomly select 500 images from the entire dataset and used the prompt "Please describe this image in detail." for the MLLM.

**Polling based Object Probing Evaluation (POPE)** Li et al. (2023b) is a VQA-based metric proposed to assess hallucinations in MLLMs. This metric evaluates the MLLM's response to the prompt "Is [object] is in this image?" To emphasize that this is a binary VQA task, we appended the prompt with "Please answer yes or no." To select objects referenced in the question prompt, we followed

three different sampling options: random, popular, and adversarial. We evaluated performance across all sampling options.

**MLLM Evaluation (MME)** Fu et al. (2023) evaluates the capabilities of MLLMs, dividing the evaluation into two major categories: perception and cognition. The perception category includes fine-grained tasks such as existence, count, location, rough color, poster, celebrity, scene, landmark, artwork identification, and OCR. The cognition category includes tasks like commonsense reasoning, numerical calculations, text translation, and code reasoning. All questions in this benchmark are structured to be answered with a simple yes or no.

Using the **LLaVA-Bench** Liu et al. (2024), we further demonstrated how well our proposed method maintains the language model performance. This benchmark involves posing various situational questions, such as dialogue, detailed descriptions, and complex reasoning, to randomly selected images from the MSCOCO val2014 dataset. A total of 60 questions are used to assess whether the model faithfully follows the instructions. The generated answers are evaluated by comparing them to the responses of a text-only GPT-4 model.

**Candidate Layers**. In dynamic premature layer selection, we partition transformer layers into buckets and select one bucket as the candidate layer set. For 32-layer LLaVA-1.5-7B, we use two buckets:[0,15),[15,31). This design limits the hyperparameter search space to only 2-4 validation runs. For efficiency, we use a validation set (MME) to select the best bucket.

## C.2 REPRODUCIBILITY

**Implementation details.** We employed greedy search as the default decoding strategy across all benchmark evaluations. For the hallucination benchmarks (POPE and CHAIR) and general-purpose benchmarks (MME, VizWiz-VQA, MMBench, MM-Vet, and LLaVA-Bench (in-the-wild)), questions from the annotation files were used as prompts, formatted to fit the chat templates of each respective MLLM. Specifically, we utilized the COCO, A-OKVQA, and GQA datasets for POPE evaluation, and MMBench_DEV_EN for MMBench. In the MM-Vet evaluation, we used an online evaluator powered by OpenAI GPT-4 to assess generated results, while for LLaVA-Bench (in-the-wild), we employed OpenAI's model gpt4-1106-preview via API. For CHAIR, a randomly sampled image set from the COCO Val2014 dataset was used across all three models, with the prompt "Please describe this image in detail." We sampled three different sets of images using different random seeds and evaluated performance by calculating the mean and standard deviation of the results. All MemVR tests were conducted using a greedy decoding approach, with do_sample=False, temperature=0, threshold=0.75, and beam=1. For VCD tests, we set do_sample=True, temperature=1, noise_step=500, and the plausibility constraint hyperparameter $\lambda$ to 0.1, while $\alpha$, which controls the degree of contrastive emphasis, was set to 1, following the default parameter settings from the original code and literature. OPERA tests were configured with beam=5, sample=True, scale_factor=50, threshold=15, num_attn_candidates=5, and penalty_weights=1. Due to OPERA's reliance on older versions of Torch and Transformers, it was incompatible with Qwen and GLM models, and thus experiments involving these models were not conducted. Additionally, our method introduces two hyperparameters: the informative layer l for activation calculations and the factor $\lambda$ to control the influence of entropy on the next token probability distribution. To map the hidden states from selected layers l to vocabulary tokens, we chose intermediate layers based on the model's depth (e.g., layers 5 to 16 for vicuna-7b, which has 32 layers), and we set $\lambda$ as a fixed value (e.g., 0.75). All parameter settings adhered to the default configurations specified in the respective papers and code repositories.

**Experimental Code.** To promote transparency and ensure the reproducibility of our work, we will release all experimental code, datasets, and detailed tutorials necessary for replicating our experiments. Our goal is to make it straightforward for researchers and practitioners to reproduce our results, regardless of their technical background. Additionally, by providing comprehensive documentation and clear guidelines, we aim to facilitate the extension of our method to other models and architectures, enabling the broader research community to explore its potential applications and improvements. We believe that open and reproducible research is essential for advancing the field and fostering collaboration.

**Computational Resources.** Our experiments were conducted on eight A40 and four A800 GPUs. The computational bottleneck was not the numerical accuracy values but the collection of potential

hallucinatory factors for analytical purposes, including logits and attention values for each head and layer.

## C.3 CASE STUDY

This case study aims to evaluate and present various benchmark cases across multiple domains systematically.

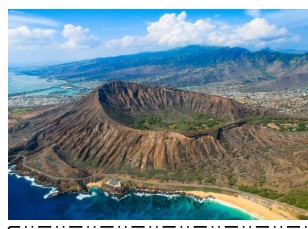

**Question:** What is the name of this famous sight in the photo?

**Ground Truth Caption:** An aerial view of Diamond Head in the Hawaiian Islands.

**Llava-v1.5-7b Default:** The famous sight in the photo is the Na Pali Coast on the island of Kaua'i in Hawaii.

**Llava-v1.5-7b + VCD:** The name of the famous sight in the photo is the Na Pali Coast of Kaua'i Island, which is an island in Hawaii.

**Llava-v1.5-7b Opera:** The name of the famous sight in the photo is the Grand Canyon.

**Llava-v1.5-7b + MemVR (Ours):** The famous sight in the photo is the Diamond Head, which is a volcanic crater located on the island of Oahu in Hawaii.

Figure 11: A case study comparing the levels of hallucination among various baselines.

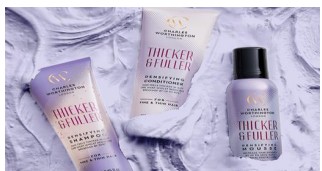

**Question:** What is the name of this famous sight in the photo?

**Ground Truth Caption:** mousse

**Llava-v1.5-7b Default:** The right most bottle is of Charles Worthington's Thicker & Fuller range.

**Llava-v1.5-7b + VCD:** densifying spray

**Llava-v1.5-7b + MemVR (Ours):** The right most bottle is the Thicker & Fuller Volumizing Mousse.

Figure 12: A case study comparing the levels of hallucination among various baselines.

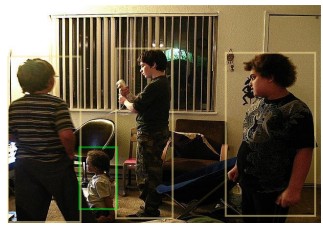

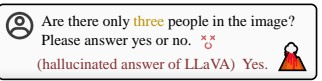

Are there only three people in the image? Please answer yes or no.

(hallucinated answer of LLaVA) Yes.

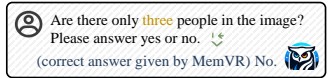

Are there only three people in the image? Please answer yes or no.

(correct answer given by MemVR) No.

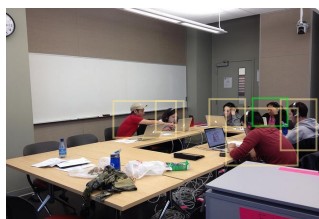

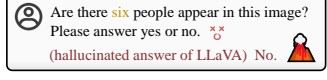

Are there six people appear in this image? Please answer yes or no.

(hallucinated answer of LLaVA) No.

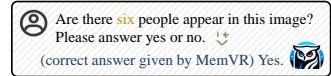

Are there six people appear in this image? Please answer yes or no.

(correct answer given by MemVR) Yes.

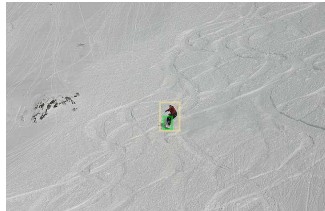

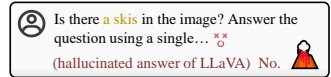

Is there a skis in the image? Answer the question using a single…

(hallucinated answer of LLaVA) No.

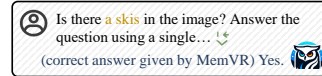

Is there a skis in the image? Answer the question using a single…

(correct answer given by MemVR) Yes.

Figure 13: A case study comparing the levels of hallucination among various baselines.

```
a = 'a dog'
b = a.replace('dog', 'cat')
print(b)
```

The image shows a python code. Is the output of the code 'a cat'? Please answer yes or no.

(hallucinated answer of LLaVA) No.

The image shows a python code. Is the output of the code 'a cat'? Please answer yes or no.

(correct answer given by MemVR) Yes.

Is this artwork created by courbet, gustave? Please answer yes or no.

(hallucinated answer of LLaVA) No.

Is this artwork created by courbet, gustave? Please answer yes or no.

(correct answer given by MemVR) Yes.

Figure 14: A case study comparing the levels of hallucination among various baselines.

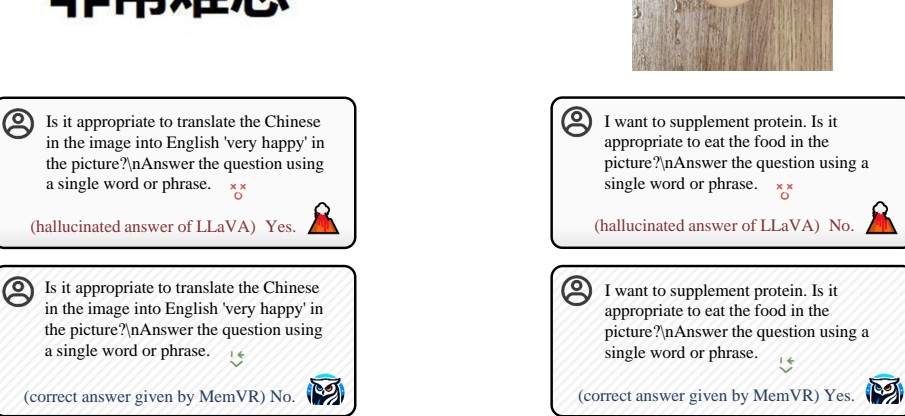

Is it appropriate to translate the Chinese in the image into English 'very happy' in the picture?\nAnswer the question using a single word or phrase.

(hallucinated answer of LLaVA) Yes.

Is it appropriate to translate the Chinese in the image into English 'very happy' in the picture?\nAnswer the question using a single word or phrase.

(correct answer given by MemVR) No.

I want to supplement protein. Is it appropriate to eat the food in the picture?\nAnswer the question using a single word or phrase.

(hallucinated answer of LLaVA) No.

I want to supplement protein. Is it appropriate to eat the food in the picture?\nAnswer the question using a single word or phrase.

(correct answer given by MemVR) Yes.

Figure 15: A case study comparing the levels of hallucination among various baselines.

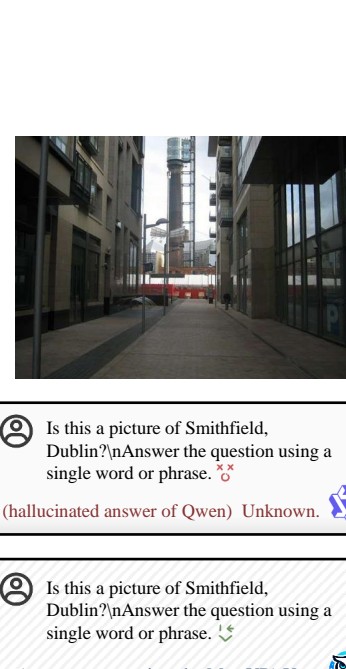

Is this a picture of Smithfield, Dublin?\nAnswer the question using a single word or phrase.

(hallucinated answer of Qwen) Unknown.

Is this a picture of Smithfield, Dublin?\nAnswer the question using a single word or phrase.

(correct answer given by MemVR) Yes.

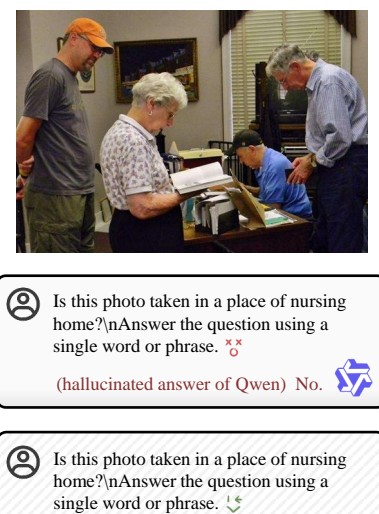
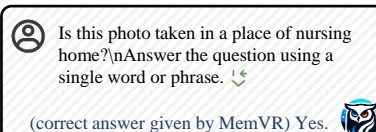

Is this photo taken in a place of nursing home?\nAnswer the question using a single word or phrase.

(hallucinated answer of Qwen) No.

Is this photo taken in a place of nursing home?\nAnswer the question using a single word or phrase.

(correct answer given by MemVR) Yes.

Figure 16: A case study comparing the levels of hallucination among various baselines.

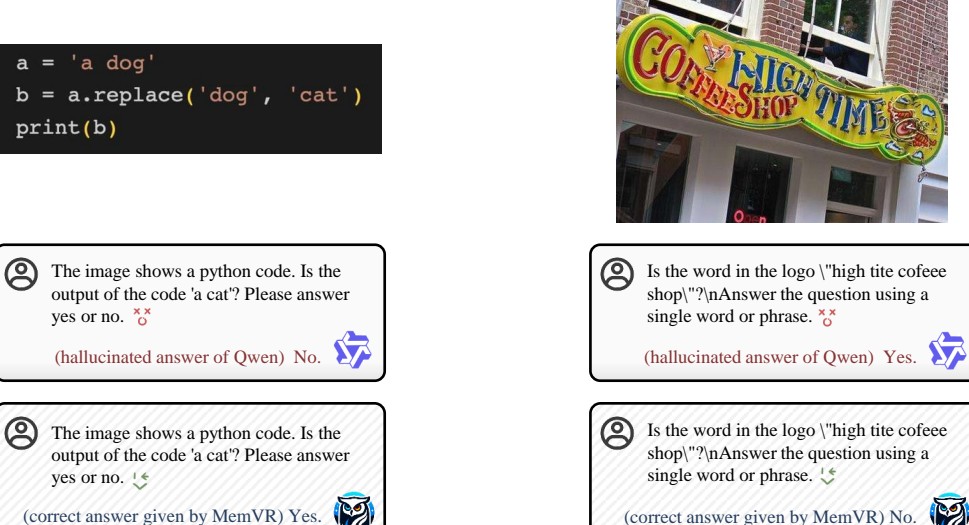

```
a = 'a dog'
b = a.replace('dog', 'cat')
print(b)
```

The image shows a python code. Is the output of the code 'a cat'? Please answer yes or no.

(hallucinated answer of Qwen) No.

The image shows a python code. Is the output of the code 'a cat'? Please answer yes or no.

(correct answer given by MemVR) Yes.

Is the word in the logo \"high tite cofeee shop\"?\nAnswer the question using a single word or phrase.

(hallucinated answer of Qwen) Yes.

Is the word in the logo \"high tite cofeee shop\"?\nAnswer the question using a single word or phrase.

(correct answer given by MemVR) No.

Figure 17: A case study comparing the levels of hallucination among various baselines.

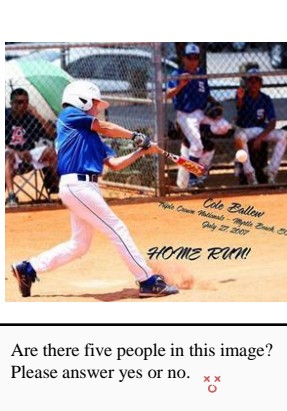

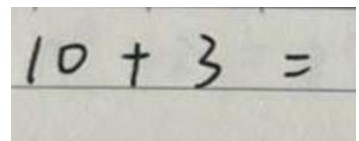

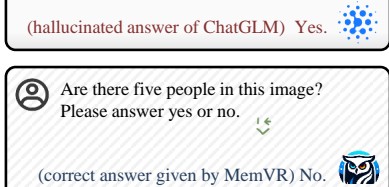

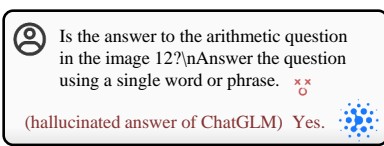

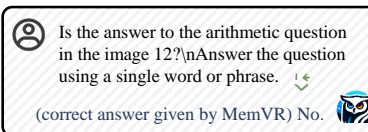

Figure 18: A case study comparing the levels of hallucination among various baselines.

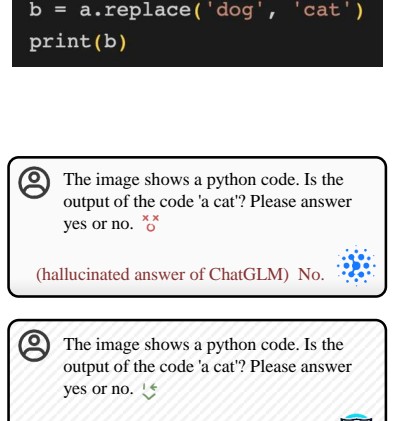

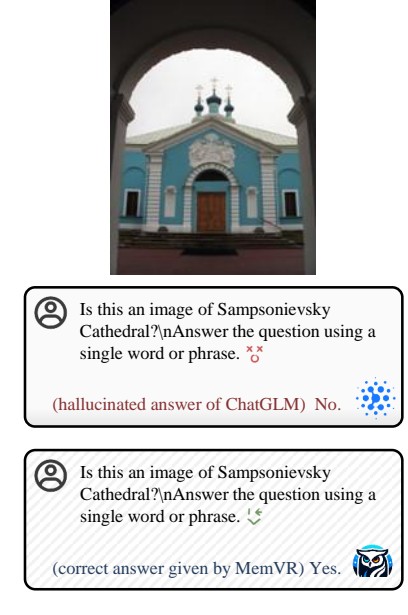

Figure 19: A case study comparing the levels of hallucination among various baselines.

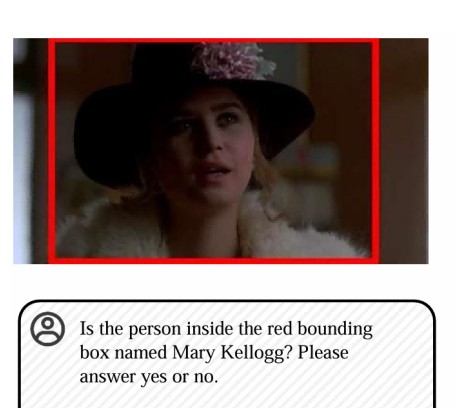
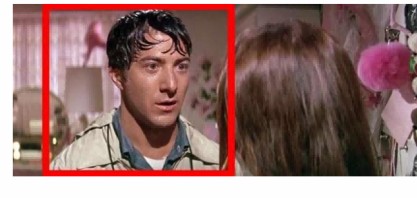
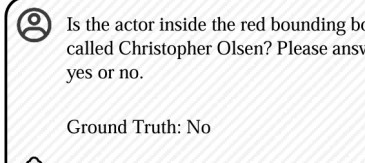
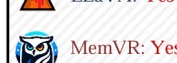
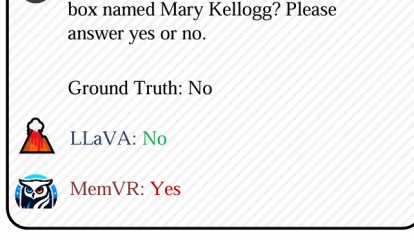

Is the person inside the red bounding box named Mary Kellogg? Please answer yes or no.

Ground Truth: No

LLaVA: No

MemVR: Yes

Is the actor inside the red bounding box called Christopher Olsen? Please answer yes or no.

Ground Truth: No

LLaVA: Yes

MemVR: Yes

Figure 20: A bad case study comparing the levels of hallucination on MME.

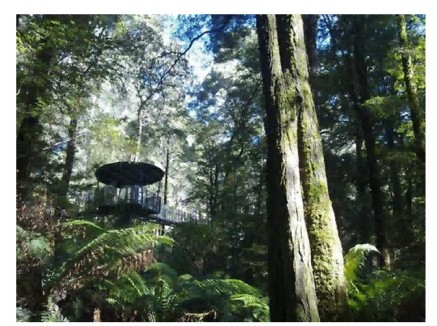
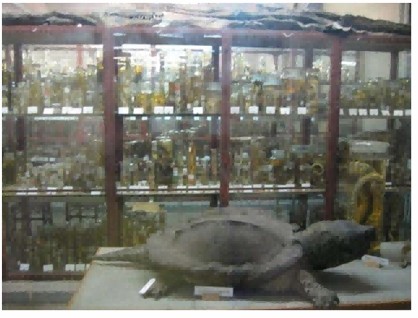
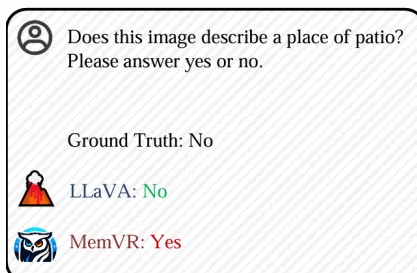
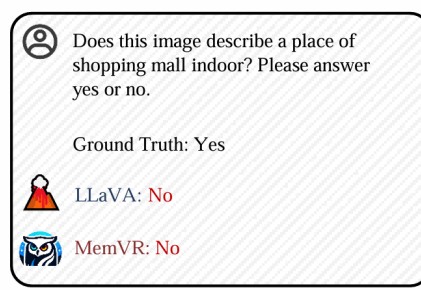

Does this image describe a place of patio? Please answer yes or no.

Ground Truth: No

LLaVA: No

MemVR: Yes

Does this image describe a place of shopping mall indoor? Please answer yes or no.

Ground Truth: Yes

LLaVA: No

MemVR: No

Figure 21: A bad case study comparing the levels of hallucination on MME.

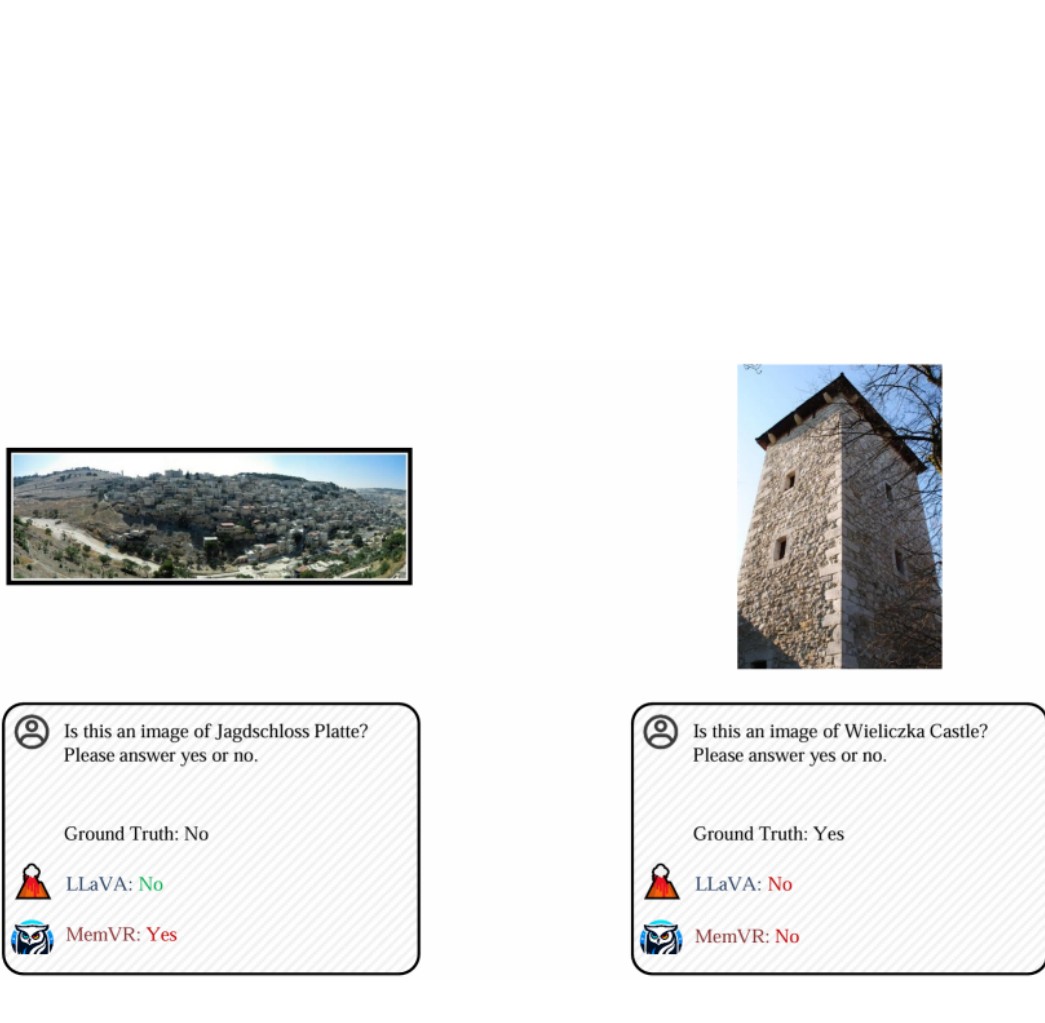

Figure 22: A bad case study comparing the levels of hallucination on MME.

## C.4 ADDITIONAL EXPERIMENTS AND RESULTS

| Strategy | 1-Token Len | 5-Token Len | 10-Token Len | 20-Token Len | 30-Token Len | 50-Token Len | 80-Token Len |
|---|---|---|---|---|---|---|---|
| Greedy | 661.7 | 897.9 | 1273.1 | 1880.3 | 2501.8 | 3617.6 | 5256.6 |
| Sample | 786.8 | 1056.2 | 1314.9 | 1998.5 | 2568.5 | 3593.0 | 5587.0 |
| VCD Sample | 1747.74 | 2767.52 | 4027.07 | 4537.42 | 5031.39 | 7690.77 | 11569.3 |
| Opera Beam | 1566.1 | 3094.9 | 4166.4 | 6242.7 | 8436.9 | 12672.3 | 19247.2 |
| MemVR Sample | 750.8 | 1197.6 | 1780.5 | 2339.2 | 2631.7 | 3718.0 | 6011.0 |
| MemVR Greedy | 775.1 | 974.2 | 1337.5 | 1861.7 | 2742.8 | 4000.9 | 5545.5 |

Table A1: Time cost for generating tokens. All based on LLaVA1.5-7B

| Method | LLaVABench (in-the-wild) | | | |
|---|---|---|---|---|
| | Average | All_1 | All_2 | All_3 |
| LLaVA1.5-7B | 64.80 ↑0.0 | 63.40 ↑0.0 | 80.20 ↑0.0 | 50.80 ↑0.0 |
| + VCD (Leng et al., 2024) | 63.20 ↓1.6 | 59.10 ↓4.3 | 82.00 ↑1.8 | 48.50 ↓2.3 |
| + OPERA (Huang et al., 2024a) | 64.30 ↓0.5 | 59.80 ↓3.6 | 83.30 ↑3.1 | 49.80 ↓1.0 |
| **+ MemVR (Ours)** | **65.17 ↑0.4** | **64.00 ↑0.6** | **80.20 ↑0.0** | **51.30 ↑0.5** |
| Qwen-VL-Chat | 68.50 ↑0.0 | 70.40 ↑0.0 | 79.30 ↑0.0 | 55.80 ↑0.0 |
| + VCD (Leng et al., 2024) | 53.77 ↓14.7 | 41.00 ↓29.4 | 85.30 ↑6.0 | 35.00 ↓20.8 |
| + OPERA (Huang et al., 2024a) | - | - | - | - |
| **+ MemVR (Ours)** | **69.50 ↑1.0** | **69.50 ↓0.9** | **82.00 ↑2.7** | **57.00 ↑1.2** |
| GLM-4V-9B | 75.30 ↑0.0 | 88.40 ↑0.0 | 73.00 ↑0.0 | 64.50 ↑0.0 |
| + VCD (Leng et al., 2024) | 74.23 ↓1.1 | 86.70 ↓1.7 | 72.80 ↓0.2 | 63.20 ↓1.3 |
| + OPERA (Huang et al., 2024a) | - | - | - | - |
| **+ MemVR (Ours)** | **76.73 ↑1.4** | **88.90 ↑0.5** | **74.80 ↑1.8** | **66.50 ↑2.0** |

Table A2: Results on LLaVABench (in-the-wild) dataset. Best-performing method per model size and dataset is highlighted in bold; arrows indicate improvement or degradation over the baseline, where higher values indicate better performance.

| Method | Total |
|---|---|
| LLaVA1.5-7B | 31.1 ↑0.0 |
| + VCD (Leng et al., 2024) | 30.20 ↓0.9 |
| + OPERA (Huang et al., 2024a) | 32 ↑0.9 |
| **+ MemVR (Ours)** | **32.4 ↑1.3** |
| Qwen-VL-Chat | 49.0 ↑0.0 |
| + VCD (Leng et al., 2024) | 34.60 ↓14.4 |
| + OPERA (Huang et al., 2024a) | - |
| **+ MemVR (Ours)** | **49.6 ↑0.6** |
| GLM-4V-9B | 63.4 ↑0.0 |
| + VCD (Leng et al., 2024) | 59.40 ↓4.0 |
| + OPERA (Huang et al., 2024a) | - |
| **+ MemVR (Ours)** | **65.0 ↑1.6** |

Table A3: Results on MM-Vet dataset. Best-performing method per model size and dataset is highlighted in bold; arrows indicate improvement or degradation over the baseline, where higher values indicate better performance.

| Method | Accuracy |
|---|---|
| LLaVA1.5-7B | 50.00 ↑0.0 |
| + VCD (Leng et al., 2024) | 44.90 ↓5.1 |
| + OPERA (Huang et al., 2024a) | 50.76 ↑0.8 |
| **+ MemVR (Ours)** | **51.50** ↑1.5 |
| Qwen-VL-Chat | 66.05 ↑0.0 |
| + VCD (Leng et al., 2024) | 34.54 ↓31.5 |
| + OPERA (Huang et al., 2024a) | - |
| **+ MemVR (Ours)** | **66.36** ↑0.3 |
| GLM-4V-9B | 57.39 ↑0.0 |
| + VCD (Leng et al., 2024) | 48.04 ↓9.4 |
| + OPERA (Huang et al., 2024a) | - |
| **+ MemVR (Ours)** | **58.00** ↑0.6 |

Table A4: Results on Vizwiz dataset. Best-performing method per model size and dataset is highlighted in bold; arrows indicate improvement or degradation over the baseline, where higher values indicate better performance.

| Method | CHAIRS | | | |
|---|---|---|---|---|
| | Cs | Ci | Recall | Len |
| LLaVA1.5-7B | 47.60 ↑0.0 | 13.30 ↑0.0 | 80.60 ↑0.0 | 99.70 ↑0.0 |
| + VCD (Leng et al., 2024) | 55.00 ↑7.4 | 15.80 ↑2.5 | 77.40 ↓3.2 | 101.20 ↑1.5 |
| + OPERA (Huang et al., 2024a) | 47.60 ↑0.0 | 13.50 ↑0.2 | 79.00 ↓1.6 | 93.20 ↓6.5 |
| **+ MemVR (Ours)** | **46.60** ↓1.0 | **13.00** ↓0.3 | 80.80 ↑0.2 | 99.60 ↓0.1 |
| GLM-4V-9B | 40.40 ↑0.0 | **9.00** ↑0.0 | 72.70 ↑0.0 | 218.20 ↑0.0 |
| + VCD (Leng et al., 2024) | 42.20 ↑1.8 | 9.60 ↑0.6 | 72.80 ↑0.1 | 239.80 ↑21.6 |
| + OPERA (Huang et al., 2024a) | - | - | - | - |
| **+ MemVR (Ours)** | **39.40** ↓1.0 | **9.00** ↑0.0 | 70.70 ↓2.0 | 214.00 ↓4.2 |
| Qwen-VL-10B | 6.80 ↑0.0 | 5.30 ↑0.0 | 53.40 ↑0.0 | 17.60 ↑0.0 |
| + VCD (Leng et al., 2024) | 13.00 ↑6.2 | 12.30 ↑7.0 | 47.90 ↓5.5 | 115.70 ↑98.1 |
| + OPERA (Huang et al., 2024a) | - | - | - | - |
| **+ MemVR (Ours)** | **4.80** ↓2.0 | **3.30** ↓2.0 | 52.30 ↓1.1 | 15.00 ↓2.6 |

Table A5: Results on CHAIRS dataset. Best-performing method per model size and dataset is highlighted in bold; arrows indicate improvement or degradation over the baseline, where lower values indicate better performance.

| Method | MMBench-Dev-EN | | | | | | |
| --- | --- | --- | --- | --- | --- | --- | --- |
| | AR | CP | FP-C | FP-S | LR | RR | Overall |
| LLaVA1.5-7B | 72.86 ↑0.0 | 75.68 ↑0.0 | 58.04 ↑0.0 | 63.48 ↑0.0 | 28.81 ↑0.0 | 51.30 ↑0.0 | 62.80 ↑0.0 |
| + VCD (Leng et al., 2024) | 60.30 | 68.58 | 51.75 | 53.24 | 18.64 | 48.70 | 54.21 |
| + OPERA (Huang et al., 2024a) | 69.85 | 75.00 | 56.64 | 66.21 | 28.81 | 53.04 | 62.80 |
| **+ MemVR (Ours)** | 71.86 ↑1.2 | 76.69 ↑1.0 | 57.34 ↓0.7 | 64.16 ↑0.9 | 31.36 ↑2.5 | 56.52 ↑5.2 | **63.75** ↑0.9 |
| GLM-4V-9B | 88.44 ↑0.0 | 86.49 ↑0.0 | 69.93 ↑0.0 | 85.67 ↑0.0 | 66.10 ↑0.0 | 85.22 ↑0.0 | 82.39 ↑0.0 |
| + VCD (Leng et al., 2024) | 86.43 | 85.47 | 68.53 | 84.64 | 61.86 | 81.74 | 80.58 |
| + OPERA (Huang et al., 2024a) | - | - | - | - | - | - | - |
| **+ MemVR (Ours)** | 88.94 ↑0.5 | 86.49 ↑0.0 | 70.63 ↑0.7 | 86.01 ↑0.4 | 66.10 ↑0.0 | 85.22 ↑0.0 | **82.65** ↑0.3 |
| Qwen-VL-10B | 60.30 ↑0.0 | 71.28 ↑0.0 | 45.45 ↑0.0 | 62.80 ↑0.0 | 28.81 ↑0.0 | 38.26 ↑0.0 | 56.53 ↑0.0 |
| + VCD (Leng et al., 2024) | 34.67 | 52.36 | 20.28 | 55.63 | 11.86 | 22.61 | 39.18 |
| + OPERA (Huang et al., 2024a) | - | - | - | - | - | - | - |
| **+ MemVR (Ours)** | 61.31 ↑1.0 | 71.28 ↑0.0 | 44.06 ↓1.4 | 62.80 ↑0.0 | 27.97 ↓0.8 | 38.26 ↑0.0 | 56.44 ↓0.1 |

Table A6: Results on MMBench dataset. Best-performing method per model size and dataset is highlighted in bold; arrows indicate improvement or degradation over the baseline, where higher values indicate better performance.

| Method | Existence | Count | Position | Color | Scene | Artwork | OCR | Numerical_cal | Text_trans | Code_reason |
| --- | --- | --- | --- | --- | --- | --- | --- | --- | --- | --- |
| LLaVA-Next (Llama3-8B) | 195.0 | 165.0 | 143.3 | 185.0 | 161.6 | 159.2 | 118.0 | 125.0 | 50.0 | 77.5 |
| +MemVR | 195.0 | 170.0 | 143.3 | 185.0 | 163.6 | 161.0 | 124.0 | 125.0 | 52.5 | 77.5 |
| LLaVA-Next (Mistral-7B) | 190.0 | 150.0 | 133.3 | 190.0 | 144.2 | 163.5 | 113.0 | 122.5 | 60.0 | 67.5 |
| +MemVR | 195.0 | 155.0 | 133.3 | 190.0 | 145.2 | 165.0 | 113.8 | 122.5 | 60.0 | 67.5 |
| LLaVA-Next (Vicuna-1.6-7B) | 195.0 | 135.0 | 143.3 | 165.0 | 162.2 | 123.2 | 132.5 | 42.5 | 107.5 | 55.0 |
| +MemVR | 195.0 | 135.0 | 135.0 | 170.0 | 163.0 | 123.5 | 140.0 | 42.5 | 115.0 | 57.5 |

Table A7: Performance comparison across different LLaVA-Next models with and without MemVR.

## C.5 SUPPLEMENT IMPLEMENT DETAIL

The code of VCD Leng et al. (2024) is also released. However, the result of VCD evaluated in our experiments (e.g. POPE and MME benchmarks) is lower than the original paper. Therefore, we report the results in the original paper.

| Method | Training-free | Hallucination Mitigation | Generalization | More modalities | Efficiency | Enhanced Components |
| --- | --- | --- | --- | --- | --- | --- |
| DoLa | ✓ | ✓ | - | ✓ | ✓ | logits |
| VCD | ✓ | ✓ | - | - | - | visual input, logits |
| OPERA | ✓ | ✓ | - | ✓ | - | attention matrix |
| HALC | ✓ | ✓ | - | - | - | visual input, logits |
| MVP | ✓ | ✓ | - | - | - | visual input, logits |
| VACoDe | ✓ | ✓ | - | - | - | visual input, logits |
| SID | ✓ | ✓ | - | - | - | text input, logits |
| API | ✓ | - | ✓ | - | - | visual input |
| AGLA | ✓ | ✓ | - | - | - | visual input, logits |
| MemVR (ours) | ✓ | ✓ | ✓ | ✓ | ✓ | **hidden states** |

Table A8: Comparison of different methods across various dimensions.

## D    EXAMPLES OF CAPABILITY INTEGRATIONS

Table A9: Six samples on MM-Vet benchmark requiring different capability integrations.

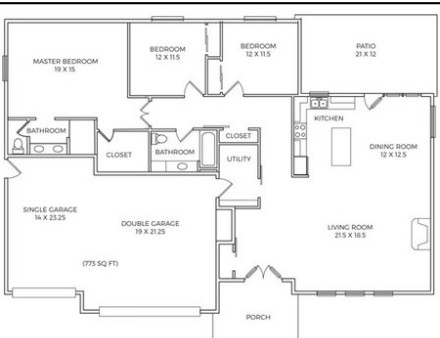

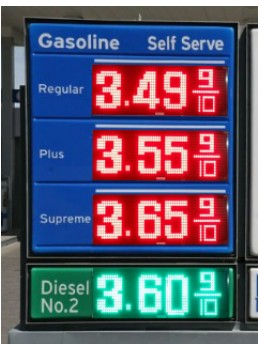

**Question:** Which room is bigger, the double garage or the living room?
**Ground Truth:** Double garage
**Required Capabilities:** OCR, Spatial Awareness, Math

**Question:** How many gallons of supreme gasoline can I get with $50?
**Ground Truth:** 13.6 | 13.7
**Required capabilities:** OCR, Math

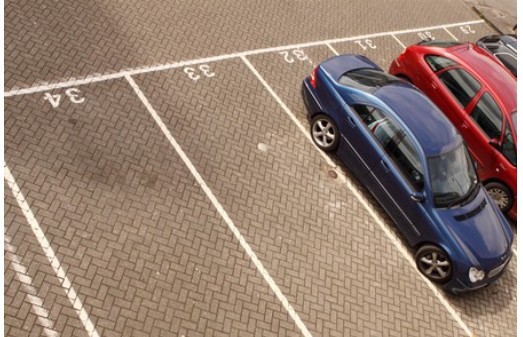

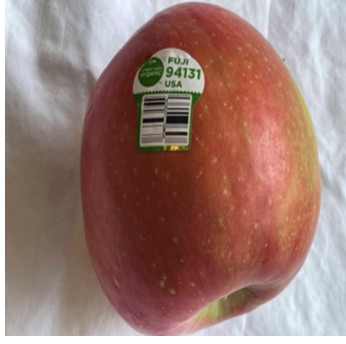

**Question:** Which car is on the parking spot 33?
**Ground Truth:** No | Empty
**Required Capabilities:** Recognition, OCR, Spatial Awareness

**Question:** Is this apple organic?
**Ground Truth:** Yes
**Required capabilities:** Recognition, OCR

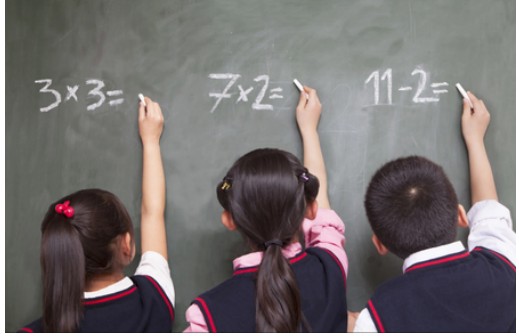

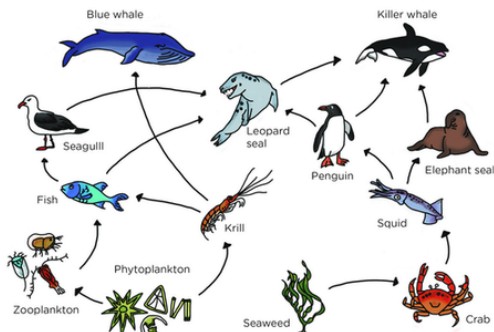

**Question:** What will the girl on the right write on the board?
**Ground Truth:** 14
**Required capabilities:** Recognition, OCR, Spatial Awareness, Math

**Question:** Which are producers in this food web?
**Ground Truth:** Phytoplankton & Seaweed
**Required Capabilities:** OCR, Knowledge, Spatial Awareness

## D.1 OTHERS

**Q: How can the model understand information directly from the vision encoder, especially if it has a different vision system?** To ensure that MEMVR is adaptable across diverse vision systems, we conducted experiments on multiple VLM architectures, including LLaVA, which utilizes a Visual-Instructional-Tuning framework with different sizes of ViT-based CLIP models, Qwen-VL-Chat, which employs a Q-Former-like architecture for visual processing, and ChatGLM-4v-9B, which integrates a large pre-trained visual encoder. These architectures encompass a broad range of vision models, providing confidence that MEMVR is applicable to most VLMs in use today.

**Artifacts and licenses** We report a list of licenses for all datasets and models used in our experiment in Table A10. We strictly follow all the model licenses and limit the scope of these models to academic research only.

| Data Sources | URL | License |
|---|---|---|
| MSCOCO 2017 | Link | CC BY 4.0 |
| ADE20K | Link | BSD-3-Clause |
| VQA Val | Link | CC BY 4.0 |
| LLaVA-bench-in-the-wild | Link | Apache-2.0 |
| ImageNet | Link | Custom License |
| MMBench | Link | Apache-2.0 |
| **Software Code** | **URL** | **License** |
| LLaVA | Link | Llama Community Licence |
| Qwen-VL | Link | Tongyi Qianwen Licence |
| GLM-4V | Link | THUDM GLM-4 Licence |
| GPT-4V/4O | Link | OpenAI Term of Use |

Table A10: License information for the scientific artifacts.

| Category | Type | Noise Step | | | | |
|---|---|---|---|---|---|---|
| | | 500 | 600 | 700 | 800 | 900 |
| Perception | Default | **1430.91** | 1338.85 | 1216.42 | 1061.55 | 897.39 |
| | Retraced | 1426.01 ↓4.90 | **1367.14** ↑28.29 | **1231.90** ↑15.48 | **1074.26** ↑12.71 | **899.64** ↑2.25 |
| Cognition | Default | 302.00 | **307.14** | **296.07** | 280.71 | 318.57 |
| | Retraced | **312.50** ↑10.50 | 307.50 ↑0.36 | 294.64 ↓1.43 | **306.43** ↑25.72 | **328.93** ↑10.36 |
| Total (P+C) | Default | 1732.91 | 1646.00 | 1512.49 | 1342.26 | 1215.97 |
| | Retraced | **1738.51** ↑5.60 | **1674.64** ↑28.64 | **1526.54** ↑14.05 | **1380.69** ↑38.43 | **1228.57** ↑12.60 |

Table A11: Mask experiment results of llava-v1.5-7b on MME, with Perception, Cognition, and Total (Per+Cog) at different noise steps. Best-performing method is highlighted in bold; arrows indicate improvement or degradation over the baseline, where higher values indicate better performance.

