# OpenReview forum: "Look Twice Before You Answer: Memory-Space Visual Retracing for Hallucination Mitigation in Multimodal Large Language Models"
_ICLR.cc/2025/Conference — Submitted to ICLR 2025_

### Official Review · Reviewer_YSEL · 2024-10-22

**Soundness:** 2
**Presentation:** 1
**Contribution:** 2
**Rating:** 3
**Confidence:** 5

**Summary:**

The paper introduces a novel approach named Memory-Space Visual Retracing (MEMVR) to address the challenge of hallucinations in Multimodal Large Language Models (MLLMs). Hallucinations in this context refer to the models' tendency to generate content that is inconsistent with the provided visual inputs, which can compromise their reliability in various applications. MEMVR is designed to mimic a human cognitive process where one would revisit visual details to seek accurate answers when initial memories fade. It operates by reinjecting visual features as "key-value memory" into the model's Feed Forward Network (FFN) layers when the model exhibits uncertainty or memory lapses regarding visual details. This method does not require external knowledge retrieval or additional fine-tuning. The paper presents comprehensive experimental evaluations demonstrating that MEMVR significantly reduces hallucination issues across various MLLMs and performs well in general benchmarks without incurring additional time overhead.

**Strengths:**

1.  MEMVR is a training-free approach that enhances the model's ability to retrieve question-relevant visual information within the FFN layers, incurring minimal additional inference overhead.
2. The method demonstrates effectiveness in various benchmarks.
3. The authors provide a thorough analysis of the relationship between model uncertainty and the tendency to hallucination.

**Weaknesses:**

1. The paper presents a method for hallucination mitigation in MLLMs by re-focusing on visual details relevant to the textual prompt. While the concept of improving model performance through prompt-related visual cues is not entirely new, with prior works like API[1], AGLA[2] exploring similar ideas. Despite some innovation in the design, the contributions do not seem to offer substantial new value or advancements in the field.
2. The paper's writing style can be overly embellished, making it less intuitive in certain parts, which may not clearly convey the intended message.
3. The phrase "signifies the other side of the coin" at line 114 is unclear and requires further explanation to understand its intended meaning.
4. While the authors claim at line 499 that MEMVR is flexible and compatible with various architectures, they also mention at line 514 that adapting the method to other structures is challenging. It raises the question of whether MEMVR can be generalized to models like LLaVA-next and InstructBlip.
5. The analysis of uncertainty in section 3.2 is based on the entropy of the decoder's output distribution, but the impact of uncertainty on the method, as shown in Figure 7, is analyzed by adding noise to the images. The claim that the method is unrelated to the biases of language priors may be questionable, as the entropy-based uncertainty measure could be connected to these biases. Additionally, it is unclear how the analysis of uncertainty in section 3.2 is directly related to the proposed method, beyond its application in the Dynamic Injection Strategy.
6. There is a typo error in line 390 where "Preception" should likely be "Perception."
7. I recommend that the authors restructure the paper, including both figures and text, to express their ideas more clearly and logically, rather than attempting to embellish the research to make it appear more sophisticated.

[1] Attention Prompting on Image for Large Vision-Language Models.

[2] AGLA: Mitigating Object Hallucinations in Large Vision-Language Models with Assembly of Global and Local Attention.

**Questions:**

See Weaknesses.

---

> ### Author Response · Authors · 2024-11-16
> **Author Response to Reviewer YSEL (Part 1/3)**
>
> Thank you for your constructive comments and evaluation of our work. We address each of your specific concerns as follows:
> ﻿
> > Q1: The paper presents a method for hallucination mitigation in MLLMs by re-focusing on visual details relevant to the textual prompt. While the concept of improving model performance through prompt-related visual cues is not entirely new, prior works like API[1], AGLA[2] exploring similar ideas. Despite some innovation in the design, the contributions do not seem to offer substantial new value or advancements in the field.
>
>
> We acknowledge the pioneering efforts in the field, including the works API[1] and AGLA[2]. Below, we highlight the unique innovations of our method across motivation, framework, efficiency, and capability.
> ﻿
>
> **1) Motivation difference**. API[1] focused on visual prompting without addressing textual queries, its motivation is not for hallucination mitigation. AGLA[2] argues that LVLMs predominantly attend to prompt-independent global image features while failing to capture prompt-relevant local features. However, we argue that hallucination may be caused by modality imbalance, i.e., LLMs struggle to perceive and memorize visual information than text, thus eliciting our idea of ''look twice before you answer'', which is grounded in a human cognitive process: *when the initial memory of certain critical visual details fades, it is intuitive to look at them for the second time to search for the accurate and hallucination-freon e answer*.
> ﻿
>
> **2) Framework difference**. API [1] uses an auxiliary model like CLIP to generate an attention heatmap for the input image dependent on the text query, then the heatmap simply multiplies the pixel values of the original image to obtain the actual input image for the LVLM. AGLA[2] uses an image-prompt matching scheme to generate an augmented view that captures local features from images, and then assembles the logits derived from both the original and augmented images. Unlike these approaches, MemVR is an architecture-agnostic, plug-and-play solution that re-injects visual features into an intermediate layer suffering from high uncertainty with the dynamic injection strategy. Moreover, MemVR also can be extended to other modalities (e.g., video, audio, 3D data), but API and AGLA cannot.
> ﻿
>
> **3) Efficiency difference**. More importantly, AP and AGLA methods introduce additional time overhead (e.g., AGLA has to reason twice to get an answer), but MemVR only with one regular inference without incurring added storage requirements and time overhead, which allows MemVR to be potentially applicable to real-world scenarios.
> ﻿
>
> **4) Capability difference**. We conduct comprehensive experiments on diverse benchmarks, including MME, POPE, CHAIR, VizWiz-VQA, MMBench, MM-Vet, and LLaVA-Bench (in-the-wild),  results demonstrate that MemVR not only significantly mitigates hallucination issues compared to SOTA methods (i.e. VCD, OPERA) across various MLLMs, but also excels in general benchmarks (notably, other approaches tend to degrade performance in general benchmarks), thus emphasizing its potential for widespread applicability.
> ﻿
>
> Here are the characteristics of different methods. We further compared HALC[3], MVP[4], VACoDe[5], SID[6].
> ﻿
> | Method | Training-free  | Viusal Hallucination Mitigation | Generalization | Extended to more modalities | Efficiency | Enhanced Components |
> | -------- | -------- | -------- | -------- | -------- | -------- | -------- |
> | DoLa | $\checkmark$ | - | - |  - | $\checkmark$ | logits |
> | VCD | $\checkmark$ | $\checkmark$ | -  |  - | - | visual input, logits |
> | OPERA | $\checkmark$ | $\checkmark$ | -  |  $\checkmark$ |  - | attention matrix |
> | HALC | $\checkmark$ | $\checkmark$ | -  |  - | - | visual input, logits |
> | MVP | $\checkmark$ | $\checkmark$ | -  |  - | - | visual input, logits |
> | VACoDe | $\checkmark$ | $\checkmark$ | - | -  | - | visual input, logits |
> | SID | $\checkmark$ | $\checkmark$ | -  |  - | - | text input, logits |
> | API | $\checkmark$ | - | $\checkmark$  |  - | - | visual input |
> | AGLA | $\checkmark$ | $\checkmark$ | -  | - |  - | visual input, logits |
> | MemVR (ours) | $\checkmark$ | $\checkmark$ | $\checkmark$ |  $\checkmark$ |  $\checkmark$ | **hidden states** |
> ﻿
>
> We have elucidated these distinctions in the revised manuscript to underscore the novelty and practical implications of our contributions.
>
> ﻿
> [1] Attention Prompting on Image for Large Vision-Language Models.
>
> [2] AGLA: Mitigating Object Hallucinations in Large Vision-Language Models with Assembly of Global and Local Attention.
>
> [3] HALC: Object Hallucination Reduction via Adaptive Focal-Contrast Decoding, ICML, 2024.
>
> [4] Look, Compare, Decide: Alleviating Hallucination in Large Vision-Language Models via Multi-View Multi-Path Reasoning.
>
> [5] Vacode: Visual augmented contrastive decoding.
>
> [6] Self-introspective decoding: Alleviating hallucinations for large vision-language models, 2024.

---

> ### Author Response · Authors · 2024-11-16
> **Author Response to Reviewer YSEL (Part 2/3)**
>
> > Q2: The paper's writing style can be overly embellished, making it less intuitive in certain parts, which may not clearly convey the intended message.
> ﻿
>
> We appreciate your observation regarding the writing style. We agree that clarity is important. We have revised the manuscript to adopt a more straightforward and concise writing style, encompassing the introduction, methodology, experiments, and conclusions, ensuring that the technical details and innovative aspects of our work are conveyed with precision and accessibility. To ensure these modifications are easily identifiable, we have distinctly marked them in blue in the updated version of our paper.
> ﻿
>
> > Q3: The phrase "signifies the other side of the coin" at line 114 is unclear and requires further explanation to understand its intended meaning.
> ﻿
>
> This expression was intended to highlight the complementary nature of our approach to existing methods. The representative methods like VCD and OPERA target to eliminate language priors, but MemVR seeks to replenish visual information. More intuitively, whether eliminating linguistic priors or supplementing visual information, the common objective is to balance the model's understanding of different modalities, which signifies the two sides of the coin. We have provided a clearer explanation in the revised manuscript to ensure that readers grasp the intended meaning and the rationale behind our methodological choices.
> ﻿
> ﻿
> > Q4: While the authors claim at line 499 that MemVR is flexible and compatible with various architectures, they also mention at line 514 that adapting the method to other structures is challenging. It raises the question of whether MemVR can be generalized to models like LLaVA-next and InstructBlip.
> ﻿
>
> ﻿First, sorry for the confusing words above. In line 499, we regard MemVR as a flexible approach as it only takes a few lines of code without modifying transformers lib, and therefore can be easily applied to other MLLMs. In line 514, where we discussed the limitations of our work, we aim to express that MemVR needs some adjustments to achieve optimal performance when applied to a brand-new model due to the difference in suitable hyperparameters across different models. ﻿We have demonstrated its effectiveness across various architectures, including LLaVA (MLP), Qwen-VL (Q-Former), and GLM-4V (MLP), as follows:
> ﻿
> | Method |  POPE (MSCOCO)  |  POPE (A-OKVQA)  |  POPE (GQA)  | MME (hallucination subset) |  MM-Vet  | Vizwiz  |  LLaVABench (in-the-wild) |
> | -------- | :--------: | :--------: | :--------: | :--------: | :--------: |  :--------: | :--------: |
> | LLaVA |  81.38  | 79.13 | 79.00 | 676.05 | 31.1 | 50.00 | 80.20 |
> | +  **MemVR**   | **87.00** | **86.21**  | **85.25** | **769.75** | **32.4** | **51.50** | 80.20 |
> | Qwen-VL  | 83.79 |  84.74 |  84.41 |  693.74 | 49.0 | 66.05 | 79.30 |
> | +  **MemVR**   | **84.07** | **86.43** | **85.69** | **758.33** | **49.6** | **66.36** | **82.00** |
> | GLM-4V | 89.87  | 89.17  | 87.01  | 840.47 | 63.4 | 57.39 | 73.00 |
> |  + **MemVR** | **90.18**  | **89.51**  | **87.37**  | **845.47** | **65.0** |  **58.00** | **74.80** |
> ﻿
>
> We are confident to say that MemVR can be easily generalized to other models like LLaVA-Next, LLaVA1.6-mistral, or others. But we'd like to discuss with you a certain case: IntructBLIP, which is the only model that does not work well with MemVR. InstructBlip uses a very special Q-Former that transforms visual information into a few tokens (often no more than 20 tokens). This makes the MemVR calculation very difficult to keep the retraced information and conduct the attention-like operation, as the core calculation of MemVR can be shortened as $\text{SiLU}(\mathbf{h}\cdot\mathbf{z}_v^{\top})\cdot\mathbf{z}_v$, where $\mathbf{z}_v \in \mathbb{R}^{n_v \times d}$ denotes visual features. Such short visual token brought by InstructBlip will lead to the fact that the retraced visual information be compressed into a $\[n_v, n_v\]$ matrix where $n_v$ is usually less than 20, while the original visual information is represented as $\[n_v, d\]$ where $d$ is much higher than $n_v$, leading to information loss. Therefore conducting MemVR calculation with IntructBLIP does not seem to have any effect on improving the quality of generated answers. We provide some benchmark evaluation results of MemVR+IntructBlip as follows,
> ﻿
> | Method |  MME (perception) |  MME (Cognition) |  POPE (COCO)  |  LLaVABench (in-the-wild)  |
> | -------- | :--------: | :--------: | :--------: | :--------: |
> | InstructBLIP |  1247.64  | 252.50 | 85.01 | 62.1 |
> | + MemVR  | 1239.20 | 260.71 | 85.41 | 61.56 |
> ﻿
>
> In all, MemVR can be easily applied to almost all open-source MLLMs and achieve substantial improvements across various benchmarks.

---

> ### Author Response · Authors · 2024-11-16
> **Author Response to Reviewer YSEL (Part 3/3)**
>
> > Q5: The analysis of uncertainty in section 3.2 is based on the entropy of the decoder's output distribution, but the impact of uncertainty on the method, as shown in Figure 7, is analyzed by adding noise to the images. The claim that the method is unrelated to the biases of language priors may be questionable, as the entropy-based uncertainty measure could be connected to these biases. Additionally, it is unclear how the analysis of uncertainty in section 3.2 is directly related to the proposed method, beyond its application in the Dynamic Injection Strategy.
> ﻿
>
> Thanks for your valuable feedback. We would like to clarify a few points regarding the concerns you've raised:
> ﻿
>
> - **Regarding the relationship between the uncertainty analysis in Section 3.2 and Figure 7:** 1) In Section 3.2, we quantify model uncertainty based on the entropy of the decoder’s output distribution. This semantic uncertainty is used to quantify the state of confusion at the different layers. 2) However, in Figure 7, we introduce noise to the visual feature when visual retracing, not the image, i.e., only affects the visual features of 'second look'. This visual uncertainty implies the cleanliness of the feature being reinjected. This experiment is intended to demonstrate that MemVR is sensitive to the quality of visual features that need to be reinjected when visual retracing, and is efficient at understanding shallow information. We have revised Figure 7 and the text to avoid confusion.
> ﻿
> - **On the relationship between our method and language prior biases:** We apologize for possibly misinterpreting you. 1) we do not claim that our approach is independent of linguistic a priori bias. 2) We claim that in contrast to previous methods, which primarily focus on eliminating biases of language priors, MemVR seeks to replenish visual information as in line 112. 3) The entropy-based uncertainty measure is used to guide our dynamic injection strategy to optimize the visual retracing process. Our experimental results also indicate that this strategy effectively enhances model performance across diverse scenarios.
> ﻿
> - **The direct connection between the uncertainty analysis in Section 3.2 and the proposed method:** The uncertainty analysis in Section 3.2 serves as a theoretical foundation for our dynamic injection strategy. By quantifying the uncertainty in the decoder’s output, we gain insight into the states of different layers across varying inputs, which guides the design of our strategy to optimize the information injection process. This analysis directly supports the core concept of our method and contributes to the improvement of overall model performance. As Reviewer eZB5 said: ''The paper provides a clear introduction to the hallucination problem in MLLMs, analyzes its causes, presents solutions, and offers a theoretical understanding of the methods used''
>
>
> > Q6: There is a typo error in line 390 where "Preception" should likely be "Perception."
> ﻿
>
> Thank you! We apologize for the typographical error and have corrected "Preception" to "Perception" in the revised manuscript.
> ﻿
>
> > Q7: I recommend that the authors restructure the paper, including both figures and text, to express their ideas more clearly and logically, rather than attempting to embellish the research to make it appear more sophisticated.
> ﻿
>
> We concur with your recommendation to restructure the paper for improved clarity and logical flow. We have revised the figures and text to better express our ideas in the updated version, focusing on clarity and conciseness. We present our research in a manner that is both rigorous and accessible to the community.
>
>
> We are committed to addressing these concerns thoroughly and believe that these revisions will significantly enhance the quality and impact of our manuscript. Thank you once again for your valuable feedback.

---

> > ### Author Response · Authors · 2024-11-22
> > **Summary of response and look forward to the feedback**
> >
> > We sincerely appreciate your constructive feedback on our manuscript. Based on your suggestions, we have made significant enhancements and clarifications, which we believe have substantially improved the quality and clarity of our work.
> > ﻿
> > - **Clarification on novelty:** We provided a detailed comparison between our approach and prior works (API, AGLA, HALC, etc.) in ''Author Response to Reviewer YSEL (Part 1/3)'', to emphasize the unique contributions of MemVR in terms of motivation, framework design, efficiency, and capability. **Notably, we are the first to alleviate hallucinations in MLLMs with only regular inference without incurring added time overhead!**
> > ﻿
> >
> > - **Improved writing clarity:** Responding to your observations, we have adopted a more concise and straightforward writing style across the manuscript, ensuring accessibility and precision. We revised unclear phrases like "signifies the other side of the coin" and distinctly marked all changes in blue in the updated manuscript.
> > ﻿
> > - **Enhanced applicability of MemVR:** We clarified the generalizability of MemVR across various architectures, showcasing its effectiveness on models like LLaVA, Qwen-VL, and GLM-4V while explaining its limitations with InstructBlip due to the model's unique characteristics. Benchmark results have been included to support our claims.
> > ﻿
> > - **Refinement of uncertainty:** We elaborated on the role of uncertainty in our method, distinguishing semantic uncertainty from visual uncertainty and their respective contributions to the proposed dynamic injection strategy. We also clarified the connection between the theoretical analysis in Section 3.2 and its application in MemVR.
> > ﻿
> > - **Typographical Corrections:** Typos such as "Preception" were corrected, and we restructured figures and text for improved logical flow and clearer presentation of ideas.
> > ﻿
> > - **Revised Figures and Analyses:** Figures were refined to avoid confusion, and additional comparisons were included to demonstrate the robustness of MemVR under diverse benchmarks, supporting its practical significance and flexibility.
> > ﻿
> > We value your feedback and have strived to thoughtfully address each concern. We look forward to your thoughts on these revisions and remain committed to further improving our work based on your insights.

---

> ### Author Response · Authors · 2024-11-25
> **We are keen to discuss further with you**
>
> Dear Reviewer YSEL,
>
> Thank you for your valuable time and the constructive feedback you have provided. We genuinely hope reviewer YSEL could kindly check our response.
>
> As the deadline for the discussion period is nearing, we would greatly appreciate it if you could kindly let us know whether there are any further questions. Thank you!
>
> Best wishes,
>
> Authors

---

> > ### Author Response · Authors · 2024-11-26
> > **We are keen to discuss further with you**
> >
> > Dear Reviewer YSEL,
> >
> > Thank you for your valuable time. In our response, we have (1) clarified the unique innovations of our method; (2) conducted additional experiments on InstructBlip; (3) clarified the meaning of "signifies the other side of the coin"; (4) clarified all other concerns; (5) revised the manuscript to adopt more straightforward writing.
> >  We genuinely hope reviewer YSEL could kindly check our response.
> >
> > As the discussion period is approaching its end, we would really appreciate it if you could kindly let us know whether there are any further questions. We will be more than happy to address them.
> >
> > Best wishes,
> >
> > Authors

---

> > > ### Author Response · Authors · 2024-11-27
> > > **Official Comment by Authors**
> > >
> > > Dear Reviewer
> > >
> > > May we kindly inquire if the provided responses have adequately addressed any questions you might have had? Does there remain a requirement for further explanations or clarifications? We wish to express our sincere gratitude for your meticulous evaluation and for generously investing a significant amount of your time in reviewing our paper. Your feedback would be greatly valued.

---

> ### Comment · Reviewer_YSEL · 2024-11-27
>
> After reviewing the updated version, I noticed that many key concepts still lack the necessary context, such as “visual prompts” mentioned in line 19, which was not sufficiently clarified as per my previous comments. Additionally, the authors have avoided providing results for their method on the LLAVA-Next model, which I had requested. Given these concerns, I will maintain my original score.

---

> ### Author Response · Authors · 2024-11-28
> **Author Response to Reviewer YSEL**
>
> > Concern1: I noticed that many key concepts still lack the necessary context, such as “visual prompts” mentioned in line 19.
> ﻿
>
> We believe that anyone working on MLLMs would be familiar with the phrase 'visual prompt', as it's one of the most basic concepts within this field. It's quite a pity that you have no idea what it stands for, we would kindly provide you with a simple explanation: visual prompts often refer to the visual (image) information that will be taken as the input of the Language Model. For the models mentioned in this paper, visual prompts are visual (image) tokens with the same dimension as embedded text tokens.
> ﻿
> ﻿
> ﻿
> > Concern2: Additionally, the authors have avoided providing results for their method on the LLAVA-Next model, which I had requested.
> ﻿
>
> To the end, we've already provided 4 MLLMs with 7 kinds of benchmark evaluations. These MLLMs used different Language models, text-image aligning methods, and training strategies to ensure effectiveness and generalization across the various testing environments. And, we have demonstrated that MemVR can be easily applied to almost all open-source MLLMs. Compared with SOTA methods, we have done the most complete experiments, as follows,
> ﻿
> ﻿
> | Method | Number of MLLM models for evaluation  | Benchmarks for evaluation |
> | -------- | -------- | -------- |
> | VCD | 3 backbones   | POPE, MME, LLaVA-Bench (3)  |
> | OPERA | 4 backbones  | CHAIR, POPE, MME, MMBench (4) |
> | HALC | 3 backbones  | POPE, MME, LLaVA-Bench (3) |
> | MemVR (ours) | 4 backbones | CHAIR, POPE, MME, MMBench,  **MM-Vet, LLaVA-Bench, VizWiz-VQA (7)** |
> ﻿
>
> We have provided 4 MLLMs across 7 kinds of benchmark evaluations, including hallucination and general benchmarks, which is sufficient to demonstrate the efficacy and generalization of our method. But you're asking for more experiments on LLaVA-next, which I would doubt for your so-called 'concerns'.
> ﻿
>
> Even though this is an extremely unreasonable request, we have decided to fulfill it. Here's the MME evaluation of LLaVA-Next (Llama3-8B) model and +MemVR:
> ﻿
>
> | Method | Existence  | Count | Position | Color | Scene | Artwork | OCR |  Numerical_calculation | Text_translation | Code_reasoning |
> | -------- | -------- | -------- |  -------- |  -------- |  -------- |  -------- |  -------- |  -------- |  -------- |  -------- |
> | LLaVA-Next (Llama3-8B) | 195.0  | 170.0  | 143.3  |  185.0   |  159.2 | 118.0   |  125.0  |  50.0  |  77.5  |  55.0  |
> | **+MemVR** | 195.0  | 170.0 | 143.3   |  185.0 |   **161.0**  | **124.0**   | 125.0   |  **52.5**  |    77.5  |  55.0  |
>
>
>
> Further, we also evaluated LLaVA-Next (Mistral-7B)  and LLaVA-Next (Vicuna-1.6-7B),
>
> | Method | Existence  | Count | Position | Color | Scene | Artwork | OCR |  Numerical_calculation | Text_translation | Code_reasoning |
> | -------- | -------- | -------- |  -------- |  -------- |  -------- |  -------- |  -------- |  -------- |  -------- |  -------- |
> | LLaVA-Next (Mistral-7B) | 190.0  | 150.0  | 133.3  |  190.0  | 163.5  |  113.0 | 122.5   |   60.0  |  67.5  |  62.5  |
> | **+MemVR** | **195.0**  | **155.0** | 133.3   |  190.0  |  **165.0**  |   **113.8**  | 122.5  |  *60.0  |  67.5  |  62.5  |
>
>
> | Method | Existence  | Count | Position | Color | Scene | Artwork | OCR |  Numerical_calculation | Text_translation | Code_reasoning |
> | -------- | -------- | -------- |  -------- |  -------- |  -------- |  -------- |  -------- |  -------- |  -------- |  -------- |
> | LLaVA-Next (Vicuna-1.6-7B) | 195.0  | 135.0  | 143.3  |  165.0  |  162.2  |  123.2 | 132.5   |   42.5  | 107.5  |  55.0  |
> | **+MemVR** | 195.0  | 135.0 | 135.0   |  **170.0**  |   **163.0**  |   **123.5**  | **140.0**   |   42.5  |  **115.0** |  **57.5** |

---

> > ### Comment · Reviewer_YSEL · 2024-11-29
> >
> > There is a clear conceptual confusion regarding the term “visual prompt” in this paper. In the context of MLLMs, “visual prompts” are typically understood as objects like bounding boxes or masks that guide the generation process [1,2,3]. This is different from the earlier understanding of “visual prompts” in pre-MLLM models [4], where they often referred to learnable parameters used for transfer learning. However, the way author define and use the term in the response does not align with either of these common interpretations, leading to further confusion. This paper does not clearly distinguish between these different uses, leading to a mixed and unclear presentation of the concept. Overall, the writing quality of the paper is extremely poor.
> >
> > [1] Visual Prompting in Multimodal Large Language Models: A Survey
> >
> > [2] Draw-and-Understand: Leveraging Visual Prompts to Enable MLLMs to Comprehend What You Want
> >
> > [3] Set-of-Mark Prompting Unleashes Extraordinary Visual Grounding in GPT-4V
> >
> > [4] Visual Prompt Tuning

---

### Official Review · Reviewer_tPgc · 2024-11-03

**Soundness:** 3
**Presentation:** 1
**Contribution:** 3
**Rating:** 5
**Confidence:** 4

**Summary:**

Despite their impressive capabilities, Multimodal Large Language Models (MLLMs) are susceptible to hallucinations, especially assertively fabricating content not present in the visual inputs. To address this issue the authors introduce a hallucination mitigation paradigm named Memory-space Visual Retracing (MEMVR). They design a dynamic premature layer injection strategy with visual retracing in MLLMs, mimicking human intuitive thinking to revisit image features for self-consistency and credible answers when pivotal memories are scrambled. Comprehensive experimental results demonstrate the effectiveness of MEMVR.

**Strengths:**

1. The paper uses concise and intuitive charts to illustrate the limitations of current methods based on Contrastive Decoding, while also clearly presenting the key design features of the proposed MEMVR framework.

2. Two statistical findings are presented that visually demonstrate the uncertainty present in large language models (LLMs) during multi-layer decoding. The authors provide initial insights into the causes of this uncertainty, adding depth to the analysis.

3. The proposed MEMVR is introduced as a training-free paradigm aimed at mitigating hallucinations in multimodal large language models (MLLMs).

**Weaknesses:**

The paper’s explanation for the occurrence of hallucinations in MLLMs is not sufficiently persuasive. While the authors identify uncertainty in the decoding process, they fail to provide adequate evidence that this uncertainty stems from the forgetting of visual features, subsequently leading to hallucinations. To strengthen this claim, I recommend that the authors conduct additional experiments, such as directly masking input image tokens to simulate the effect of forgetting. They could then test whether MEMVR is able to recover visual features through the "look twice" mechanism while utilizing the normal image tokens during the recovery process.

Poor Presentation. The presentation of the proposed method is convoluted and lacks clarity. The authors unnecessarily complicate straightforward concepts by introducing numerous "unusual" symbols and formulas. While this may create an impression of sophistication, it significantly increases the reading difficulty for the audience. For instance, in Equation (2), the symbol
 typically denotes proportionality; however, the intended meaning remains unclear. Furthermore, the relationship between the content of Equation (2) and its explanation in Equation (3) is separated by a substantial amount of text, making it laborious for readers to follow. Additionally, the terms
 and
 mentioned in line 256 lack proper definitions, leading to confusion—if they represent trainable parameters of the FFN, it is unclear why they are treated as keys and values. There are multiple instances of such issues throughout the manuscript, indicating that it is not a well-presented piece of work. Here are some suggested modifications:
1. Provide a clear definition and explanation for each symbol when it is first introduced, especially for non-standard usage like $\propto$ in Equation (2).
2. Reorganizing the text to keep related equations and their explanations closer together, particularly for Equations (2) and (3).
3. Clearly define the terms $k$ and $v$ and explain the rationale behind treating them as keys and values if they are indeed FFN parameters.
4. Recommending the inclusion of a notation table or glossary to help readers keep track of symbols and their meanings throughout the paper.
5. Try to review the entire manuscript to identify and simplify overly complex formulations without losing essential information.

**Questions:**

See weakness.

---

> ### Author Response · Authors · 2024-11-16
> **Author Response to Reviewer tPgc  (Part 1/2)**
>
> Thank you for your insightful comments. We address your concerns as follows.
>
> > Q1:  The paper’s explanation for the occurrence of hallucinations in MLLMs is not sufficiently persuasive. While the authors identify uncertainty in the decoding process, they fail to provide adequate evidence that this uncertainty stems from the forgetting of visual features, subsequently leading to hallucinations. To strengthen this claim, I recommend that the authors conduct additional experiments, such as directly masking input image tokens to simulate the effect of forgetting. They could then test whether MEMVR is able to recover visual features through the "look twice" mechanism while utilizing the normal image tokens during the recovery process.
>
>
> Thank you for this valuable suggestion. Following your recommendation, we conducted additional experiments by masking input images with randomly generated noise. These masked images were then used as input to the VLM, allowing us to test the ability of MemVR to retrace features from the unmasked image through the "look twice" mechanism. We experimented with varying levels of noise, where higher levels simulate increased forgetting of the original image. At 900 noise steps, the human eye can no longer distinguish the content of the picture.
>
> Mask experiment results of llava-v1.5-7b on MME, with Perception, Cognition, and Total (Per+Cog) scores at different noise steps as follows,
>
> Perception:
>
> | Method \ Noise step |  500 | 600  | 700 | 800 | 900 |
> | :----------: | :----------: | :----------: | :----------: | :----------: | :----------: |
> | Default  | 1430.91| 1338.85| 1216.42| 1061.55| 897.39 |
> | Retraced (look twice) | 1426.01| 1367.14| 1231.90| 1074.26| 899.64 |
>
> Cognition:
>
> | Method \ Noise step |  500 | 600  | 700 | 800 | 900 |
> | :----------: | :----------: | :----------: | :----------: | :----------: | :----------: |
> | Default  | 302.00 | 307.14 | 296.07 | 280.71 | 318.57 |
> | Retraced (look twice) |  312.50 | 307.50 | 294.64 | 306.43 | 328.93 |
>
> Total (P+C):
>
> | Method \ Noise step |  500 | 600  | 700 | 800 | 900 |
> | :----------: | :----------: | :----------: | :----------: | :----------: | :----------: |
> | Default  | 1732.91| 1646.00| 1512.49| 1342.26| 1215.97|
> | Retraced (look twice) |  1738.51| 1674.64| 1526.54| 1380.69| 1228.57 |
>
>
> As shown in the above tables, MemVR is able to recover some of visual memories through the 'look twice' mechanism under different noise steps, enhancing performance. The experiments demonstrate that retracing ('look twice') using the unmasked image indeed enhances the MLLM's understanding of the masked inputs. We believe this result offers strong evidence for the effectiveness of the "look twice" mechanism, adding further rigor to the logical framework of our paper. Thank you for this valuable suggestion, which has made a meaningful contribution to our work.

---

> ### Author Response · Authors · 2024-11-16
> **Author Response to Reviewer tPgc  (Part 2/2)**
>
> > Q2: The presentation of the proposed method is convoluted and lacks clarity. The authors unnecessarily complicate straightforward concepts by introducing numerous "unusual" symbols and formulas. [...]
>
>
> Sorry for the unclear presentation of MemVR. To address your concern, we have provided a clearer explanation in the revised manuscript to ensure that readers understand our methodology.
>
> Here's a detailed explanation of how MemVR is implemented.  In brief, as the title "Look twice before you answer" suggests, this is the main process of our MemVR. Specifically, this involves two issues: 1) at which layer to re-look, and 2) how to re-look at it.
>
> For the first issue - at which layer to re-look, we design a dynamic strategy to select a specific injection layer for 're-look', i.e.,  replenish visual information in the selected layer. In particular, Algorithm 1 in line 277 shows the code flow of our dynamic injection strategy. We first compute the uncertainty of the next token probability on each early layer and set the threshold $\gamma$. When the uncertainty of a specific layer is over $\gamma$, 're-look' is triggered in this layer.
>
> Then, how to 're-look' at it, i.e., how visual retracing is implemented. We take the visual feature after the connector as visual evidence to be re-injected at the trigger layer selected in the previous step. Taking the hidden state $x \in \mathbb{R}^{d}$ of the trigger layer as query, and the visual evidence $z_v\in \mathbb{R}^{n_v \times d}$ as both key and value, then we perform a similar cross-attention operation, i.e, $\operatorname{Retrace}({z}_v \mid {x})=\phi({x} \times {z}_v^\top) \times {z}_v$. Further, in FFN, we re-inject the retrieved visual memory to the hidden-state with ${\operatorname{FFN}}({x} \propto z_v)=(1-\alpha)\operatorname{FFN}(x) + \alpha\operatorname{Retrace}({z}_v \mid {x})$, where $\alpha$ denotes the injection ratio of visual memory.
>
>
> That's the whole process of our method, simple, but efficient and effective. We have revised the specific process of MemVR in the revised manuscript to make our method more clear and accessible. Please do let us know if you have any questions or concerns on it.
> ﻿
>
> > Here are some suggested modifications: 1.Provide a clear definition and explanation for each symbol when it is first introduced, especially for non-standard usage like in Equation (2). 2.Reorganizing the text to keep related equations and their explanations closer together, particularly for Equations (2) and (3). 3.Clearly define the terms $k$ and $v$ and explain the rationale behind treating them as keys and values if they are indeed FFN parameters. 4.Recommending the inclusion of a notation table or glossary to help readers keep track of symbols and their meanings throughout the paper. 5.Try to review the entire manuscript to identify and simplify overly complex formulations without losing essential information.
>
> Thanks for your valuable suggestions. We have provided a clearer explanation and definition for each symbol, simplified overly complex formulations, and reorganized the text in the revised manuscript to ensure that readers understand our methodology. These modifications are highlighted in blue in the revised version of our paper to facilitate easy identification. We appreciate your guidance in improving the clarity of our discussion.

---

> > ### Author Response · Authors · 2024-11-22
> > **Summary of response and look forward to the feedback**
> >
> > We greatly appreciate the thoughtful critique and suggestions from Reviewer tPgc. Below is a summary of our revisions and clarifications based on the provided feedback.
> > ﻿
> > - **Clarification and Evidence Supporting "Look Twice" Mechanism:** Following your insightful suggestion, we conducted additional experiments by masking input images with noise to simulate forgetting visual features. The results demonstrate MemVR’s ability to recover visual information through the "look twice" mechanism, validating its effectiveness in mitigating hallucinations caused by forgotten visual details. These results strengthen the logical framework of our work.
> > ﻿
> > - **Simplified and Clearer Methodology Explanation:** We restructured the presentation of MemVR to improve clarity. The revised manuscript now provides a detailed yet straightforward explanation of the dynamic injection strategy and visual retracing process, addressing concerns regarding convoluted symbols and formulas. We also reorganized related equations and their explanations for better coherence.
> > ﻿
> > - **Incorporation of Suggested Modifications:** We incorporated all specific suggestions, including providing clear explanations for symbols like $k$ and $v$, improving the logical flow of text.
> > ﻿
> > We hope these revisions and clarifications address your concerns and look forward to any additional feedback or questions.

---

> ### Author Response · Authors · 2024-11-25
> **We are keen to discuss further with you**
>
> Dear Reviewer tPgc,
>
> Thank you for your valuable time and the constructive feedback you have provided. We genuinely hope reviewer tPgc could kindly check our response.
>
> As the deadline for the discussion period is nearing, we would greatly appreciate it if you could kindly let us know whether there are any further questions. Thank you!
>
> Best wishes,
>
> Authors

---

> > ### Comment · Reviewer_tPgc · 2024-12-02
> >
> > Thank you to the authors for conducting the supplementary experiments on "Look Twice" and revising the manuscript. The experimental results convincingly demonstrate that as noise levels increase, the uncertainty in large language model (LLM) responses also rises, effectively capturing the phenomenon of LLMs "forgetting" visual features during reasoning. However, the "Look Twice" retrace appears to have limited efficacy in mitigating this issue, as the observed performance improvements are minimal. In summary, while the authors successfully model the phenomenon of visual feature "forgetting," the "Look Twice" retrace does not achieve satisfactory performance levels in the context of image annotation.
> >
> > The revised manuscript is commendably clearer and more concise compared to its initial version. Nevertheless, a notable disconnect persists between the textual explanation and the figures. While the figures effectively convey the study's motivation and core ideas, the accompanying text lacks sufficient detail and intuitive explanations to complement the visual representations.
> >
> > I thank the authors for the detailed response and appreciate your patience.

---

> > > ### Author Response · Authors · 2024-12-02
> > > **Official Comment by Authors**
> > >
> > > Dear Reviewer tPgc,
> > >
> > > Glad to hear from you. We value your opinions, and we’ll for sure make our paper more intuitive.
> > >
> > > We have demonstrated its effectiveness across various benchmarks, as follows:
> > >
> > > | Method |  POPE (MSCOCO)  |  POPE (A-OKVQA)  |  POPE (GQA)  | MME (hallucination subset) |  MM-Vet  | Vizwiz  |  LLaVABench (in-the-wild) |  CHAIR |
> > > | -------- | :--------: | :--------: | :--------: | :--------: | :--------: |  :--------: | :--------: |  :--------: |
> > > | LLaVA |  81.38  | 79.13 | 79.00 | 676.05 | 31.1 | 50.00 | 58.8 | 77.1 |
> > > | + VCD |  84.66  | 80.99 | 81.74 | 717.52 | 30.2 | 44.90 | 57.8 | 77.3 |
> > > | + OPERA  |  84.77  | 84.27 | 84.03 | 708.04 | 32.0 | 50.76 | 59.5 | 76.8 |
> > > | +  **MemVR**   | **87.00** ↑5.62 | **86.21** ↑7.08 | **85.25** ↑6.25 | **769.75** ↑93.70 | **32.4** ↑1.3 | **51.50** ↑1.50 | **63.8** ↑5.0 | **80.8** ↑3.7|
> > >
> > > Here is the inference time (ms) of different methods:
> > >
> > > | Method   | 20-Token Len | 50-Token Len | 80-Token Len |
> > > |----------|--------------|--------------|--------------|
> > > | LLaVA-1.5| 1880.3 | 3617.6 | 5256.6  |
> > > | +VCD     | 4537.4 ↑x2.4 | 7690.8 ↑x2.1 | 11569.3 ↑x2.2|
> > > | +OPERA   | 6242.7 ↑x3.3 | 12672.3 ↑x3.5| 19247.2 ↑x3.7|
> > > | +MEMVR   | 1861.7 ↑x1.0 | 4000.9 ↑x1.0 | 5545.5 ↑x1.0 |
> > >
> > > Compared with several approaches that are also training-free, it turns out that MemVR **achieves both the highest efficiency and performance.**
> > > ﻿
> > > MemVR even outperforms LLaVA-Next vicuna1.5-7b and vicuna1.5-13b in MME total score with base model of LLaVA1.5 vicuna-7b, where you may notice that scaling up the Language Model of LLaVA-Next vicuna1.5 from 7b to 13b improves the score by 50, and yet MemVR, as a training-free method, brings 32 of improvement to LLaVA1.5, and 36, 10 for Qwen-VL-Chat and GLM-4v-9b respectively.
> > >
> > > **Such improvements should not be considered as ‘limited efficacy’**. Furthermore, you may refer to Section C.4 in the Appendix of our paper for how MemVR outperforms VCD and OPERA across diverse benchmarks.
> > > ﻿
> > > We would like to address your concern about our method’s limited performance and thank you for your patient response.
> > >
> > > Best wishes,
> > >
> > > Authors

---

### Official Review · Reviewer_iT3w · 2024-11-04

**Soundness:** 3
**Presentation:** 2
**Contribution:** 3
**Rating:** 5
**Confidence:** 3

**Summary:**

This paper proposes a hallucination mitigation technique called MEMVR, Memory-space Visual Retracing for MLLMs. The approach reinjects visual tokens as “key-value memory” into intermediate layers when high uncertainty is detected, to reinforce alignment between visual and textual information without additional fine-tuning or external data. The author used experimental results to show that MEMVR improves hallucination mitigation across various benchmarks, and the method operates efficiently by only triggering visual reinjection when needed.

**Strengths:**

- Originality: MEMVR’s approach to hallucination mitigation is innovative, leveraging visual token reinjection without additional training or external databases. This is a straightforward solution that addresses a critical issue in multimodal models.

- Efficiency: The proposed method minimizes computational overhead by using a single inference pass and only invoking reinjection when needed, making MEMVR potentially applicable to real-world scenarios.

- Empirical Performance: MEMVR demonstrates strong results across multiple benchmarks, including improvements over Visual Contrastive Decoding (VCD) and OPERA, showing its effectiveness in reducing hallucinations and enhancing model alignment with visual content.

**Weaknesses:**

- Dependence on Threshold-based Mechanism: MEMVR's use of an uncertainty threshold for reinjection may lead to sensitivity to specific datasets or model architectures, potentially affecting its generalizability. The adaptability of this mechanism across different models without manual tuning remains uncertain.

- Section C.3 Analysis: Section C.3 highlights only the successes of MEMVR. Including instances of failure would provide a more balanced analysis. Without external data as used by other methods, MEMVR might risk reinforcing biases in the original model by reinjecting similar visual features. Addressing these issues and examining cases where MEMVR underperforms would offer a more comprehensive understanding of its limitations and inform future improvements.

- Presentation Style: The Q&A format in Section 4.2 on quantitative results lacks logical structure and clarity. It would be more effective to present this section in a structured format that clearly outlines the key findings, for example: “Q2: How well does MEMVR perform on general-purpose benchmarks?” Can be simply phrase as “MEMVR performance on general-purpose benchmarks.”

- The paper provides valuable insights but needs more attention to detail in presenting results. Specific issues include:
In Table 2, Qwen-VL-10B + OPERA's best F1 score on MSCOCO is not bolded.
Table 7 shows LLaVA1.5-7B + OPERA and LLaVA1.5-7B + VCD have top scores on gnkwrec and ocrgnsp, respectively, but these are not bolded. Additionally, a value for LLaVA1.5-7B + OPERA is listed as +1.4 instead of +20.
Table 6 reports MemVR’s improvement on Convs as +0.5 instead of +5.0.

**Questions:**

- Could the authors clarify how the uncertainty threshold was selected, and how sensitive MEMVR is to this parameter across different datasets?

- Are there specific scenarios where MEMVR fails to reduce hallucinations, could you provide a few concrete examples of failure cases and limitation analysis on them?

- Would MEMVR function effectively if extended to other modalities (e.g., audio, 3D data)? Or would it require significant adjustments?

---

> ### Author Response · Authors · 2024-11-16
> **Author Response to Reviewer iT3w (Part 1/3)**
>
> Thank you for your insightful comments. We address your concerns as follows.
>
> > Concern 1: Dependence on Threshold-based Mechanism: MemVR's use of an uncertainty threshold for reinjection may lead to sensitivity to specific datasets or model architectures, potentially affecting its generalizability. The adaptability of this mechanism across different models without manual tuning remains uncertain.
> ﻿
> ﻿
>
> Thank you for pointing out that. MemVR does need a few rounds of threshold adjustments to achieve optimal performance when applied to a brand-new model, this process is typically brief and does not present significant overhead (usually no longer than 2 hours). More importantly, in most cases, even a randomly chosen parameter could improve the performance compared to the baseline, indicating a degree of robustness. Once the optimal threshold (usually set at 0.75) is determined, MemVR achieves significant performance gains across a wide range of benchmarks, showcasing its adaptability and effectiveness in diverse settings.  On Part 2/3 Q1, we provide performance results for LLaVA under the different thresholds across three benchmarks, and the results demonstrate that MemVR is not sensitive to the threshold.
> ﻿
> > Concern 2: Section C.3 Analysis: Section C.3 highlights only the successes of MemVR. Including instances of failure would provide a more balanced analysis. Without external data as used by other methods, MemVR might risk reinforcing biases in the original model by reinjecting similar visual features. Addressing these issues and examining cases where MemVR underperforms would offer a more comprehensive understanding of its limitations and inform future improvements.
> ﻿
>
> Thank you for your valuable suggestions!  We appreciate the importance of providing a balanced analysis and found your feedback very helpful.  Specifically, we are keen to explore how reinjecting similar visual features without external data might affect model biases.  To address this, we have collected failure cases from the MME benchmark, particularly in the 'Celebrity,' 'Scene,' and 'Landmark' sub-tasks, where MemVR underperforms compared to the default model.
>
> | Right numbers | existence | count | position | color | posters | celebrity | scene | landmark | artwork | OCR | CommR | numerical_cal | translation | code |
> | :-------: | :-------: | :-------: | :-------: | :-------: | :-------: | :-------: | :-------: | :-------: | :-------: | :-------: | :-----------------------: | :-----------------------: | :-----------------: | :---------------: |
> |   Total  | 60        | 60    | 60       | 60    | 294     | 340       | 400   | 400      | 400     | 40  | 140                   | 40                    | 40              | 40            |
> |   LLaVA1.5-7B   | 58        | 51    | 45       | 54    | 241     | 266       | 342   | 352      | 286     | 32  | 97                    | 18                    | 27              | 21            |
> |   LLaVA1.5-7B + MemVR   | 58        | 51    | 46       | 54    | 241     | 264       | 341   | 351      | 288     | 32  | 102                   | 18                    | 28              | 23            |
>
>
> We categorize MemVR's failures into two types:
>
> (a) Cases where the default model provides the correct answer, but MemVR outputs an incorrect one.
>
> (b) Cases where both the default model and MemVR produce incorrect answers.
>
>
> For (a), we attribute the failure to over-disturbance of the default model's reasoning process.  In these instances, the original visual features are sufficient for reasoning, and the reinjected tokens inadvertently disrupt this process, leading to errors.  We are actively investigating methods to mitigate such disturbances.
>
>
> For (b), the failures arise from either the excessive complexity of the image or gaps in the LLM's knowledge base, which prevents correct reasoning even after retracing.
>
>
> These failure cases have been added to Section C.3 in the revised version of the paper.  We hope this analysis addresses your concerns and provides a more comprehensive understanding of MemVR's limitations, paving the way for future improvements.
> ﻿
> > Concern 3: Presentation Style: The Q&A format in Section 4.2 on quantitative results lacks logical structure and clarity. It would be more effective to present this section in a structured format that clearly outlines the key findings, for example: “Q2: How well does MemVR perform on general-purpose benchmarks?” Can be simply phrased as “MemVR performance on general-purpose benchmarks.”
> ﻿
>
> ﻿Thank you! We have re-paraphrased this section to make it easier to read and more logical. These modifications are highlighted in blue in the revised version of our paper to facilitate easy identification.

---

> ### Author Response · Authors · 2024-11-16
> **Author Response to Reviewer iT3w (Part 2/3)**
>
> > Concern 4: The paper provides valuable insights but needs more attention to detail in presenting results. Specific issues include: In Table 2, Qwen-VL-10B + OPERA's best F1 score on MSCOCO is not bolded. Table 7 shows LLaVA1.5-7B + OPERA and LLaVA1.5-7B + VCD have top scores on gnkwrec and ocrgnsp, respectively, but these are not bolded. Additionally, a value for LLaVA1.5-7B + OPERA is listed as +1.4 instead of +20. Table 6 reports MemVR’s improvement on Convs as +0.5 instead of +5.0.
> ﻿
>
> Thank you for your detailed review! We have checked all tables and in-line data to ensure these mistakes are all fixed.
>
>
> > Q1: Could the authors clarify how the uncertainty threshold was selected, and how sensitive MemVR is to this parameter across different datasets?
>
>
> Sure, when running MemVR calculations, we first calculate the early layer entropy by analyzing its early-exit logits, and we make MemVR only to be triggered when a specific layer entropy exceeds the threshold, where the threshold is determined by a set of test experiments on the MME benchmark that reveal the average uncertainty (which can also be regarded as semantic entropy) of early layers is over 0.75 when hallucination occurs. That's the reason the threshold is set to 0.75. We set the threshold to be fixed, which effectively improves performance in all benchmark tests.  For example, we set the threshold to 0.75 in LLaVA-1.5 on 9 benchmarks across different domains,  MemVR achieves dynamic injection and stable performance improvement in all of them.
>
> To further address your concern, we have now provided performance results for LLaVA under the different threshold across three benchmarks,
>
> Threshold sensitivity experiments of LLaVA-1.5+MemVR on MME:
>
> | Threshhold	| 0.50 | 0.55	| 0.60	| 0.65	| 0.70	| 0.75	| 0.80	| 0.85	| 0.90	| 0.950	| 1.00 |
> | :--------: | :--------: | :--------: | :--------: | :--------: | :--------: | :--------: | :--------: | :--------: | :--------: | :--------: | :--------: |
> | Overall	| 1851.08	| 1856.08	| 1865.33	| 1862.94	| 1878.50	| 1896.72	| 1874.76	| 1865.97	| 1853.58	| 1867.52	| 1864.68 |
> | Preception	| 1498.58	| 1503.58		| 1512.83	 | 1510.44	 | 1512.78	 | 1512.80	 | 1512.97	 | 1503.47	 | 1493.22	 | 1508.95	 | 1508.97 |
> | Cognition	| 352.50	 | 352.50	 | 352.50	 | 352.50	 | 365.71	 | 383.92	 | 361.79	 | 362.50	 | 360.36	 | 358.57	 | 355.71 |
> ﻿
>
> Threshold sensitivity experiments of LLaVA-1.5+MemVR on POPE (COCO):
>
> | Threshhold	| 0.50 | 0.55	| 0.60	| 0.65	| 0.70	| 0.75	| 0.80	| 0.85	| 0.90	| 0.950	| 1.00 |
> | :--------: | :--------: | :--------: | :--------: | :--------: | :--------: | :--------: | :--------: | :--------: | :--------: | :--------: | :--------: |
> | popular    | 87.13 | 87.07 | 87.07 | 87.07 | 87.07 | 87.30 | 87.13 | 87.30 | 87.30 | 87.30 | 87.07 |
> | random     | 88.40 | 88.37 | 88.37 | 88.37 | 88.40 | 88.47 | 88.43 | 88.43 | 88.47 | 88.47 | 88.33 |
> | adversarial| 85.17 | 85.17 | 85.17 | 85.17 | 85.17 | 85.20 | 85.17 | 85.20 | 85.17 | 85.17 | 84.87 |
> | Average    | 86.90 | 86.87 | 86.87 | 86.87 | 86.88 | 86.99 | 86.91 | 86.98 | 86.98 | 86.98 | 86.76 |
>
>
> Threshold sensitivity experiments of LLaVA-1.5+MemVR on MMBench_DEV_EN:
>
> | Threshhold	| 0.50 | 0.55	| 0.60	| 0.65	| 0.70	| 0.75	| 0.80	| 0.85	| 0.90	| 0.950	| 1.00 |
> | :--------: | :--------: | :--------: | :--------: | :--------: | :--------: | :--------: | :--------: | :--------: | :--------: | :--------: | :--------: |
> | Overall | 63.32 | 63.32 | 63.32 | 63.32 | 63.32 | 63.75 | 63.32 | 63.32 | 63.32 | 63.06 | 62.80 |
>
> From the results of the sensitivity experiments, it can be seen that MemVR is not sensitive to this parameter across different datasets. For further analysis, it can be seen that if the threshold is set too low, the dynamic injection strategy becomes static fixed-layer injection, while setting the threshold too high makes MemVR difficult to trigger and degrades into the vanilla model.

---

> ### Author Response · Authors · 2024-11-16
> **Author Response to Reviewer iT3w (Part 3/3)**
>
> > Q2: Are there specific scenarios where MemVR fails to reduce hallucinations, could you provide a few concrete examples of failure cases and limitation analysis on them?
>
>
> Sure. Please refer to the reply for Concer2 above, and we've also updated some failure cases to C.3.
>
> Additionally, we’d like to mention another failed case with MemVR. We've tested several models and almost all of them received significant improvements, except InstructBLIP, which is the only model that does not work well with MemVR. InstructBlip uses a very special Q-Former that transforms visual information into a few tokens (often no more than 20 tokens). This makes the MemVR calculation very difficult to keep the retraced information and conduct attention-like operation, as the core calculation of MemVR can be shortened as, $\operatorname{Retrace}({z}_v \mid {x})=\phi({x} \times {z}_v^\top) \times {z}_v$. Such short visual token brought by InstructBlip will lead to the fact that the retraced information will be compressed into a [$n_v, n_v$] matrix where $n_v$ is usually less than 20, while the original visual information is represented as [$n_v, d$] where $d$ is much higher than $n_v$. Therefore conducting MemVR calculation with InstructBLIP does not seem to have any effect on improving the quality of generated answers. We consider its poor performance with MemVR due to using blip2-like Q-Former architecture, which could lead to serious information loss.
>
>
>
> We provide some benchmark evaluation results of MemVR+IntructBlip as follows,
>
> | Method |  MME (perception) |  MME (Cognition) |  POPE (COCO)  |  LLaVABench (in-the-wild)  |
> | -------- | :--------: | :--------: | :--------: | :--------: |
> | InstructBLIP |  1247.64  | 252.50 | 85.01 | 62.1 |
> | + MemVR  | 1239.20 | 260.71 | 85.41 | 61.56 |
>
> Note: Please refer to C.2, Implementation details for experimental settings.
>
> Please do let us know if you have any questions or concerns on it.
>
> > Q3: Would MemVR function effectively if extended to other modalities (e.g., audio, 3D data)? Or would it require significant adjustments?
>
>
> The core contribution of MemVR is 'retracing visual information at the second time'.    Following this idea, there are two questions when extending it to more modalities
>
> 1) How's the alignment work going? Can the large language model fully understand the transformed token from, e.g., audio or 3D mesh or something else?    If yes, it means that you now have a powerful alignment tool, like efficient projection MLPs.  In this way, we can ensure that the model can, somehow, understand the tokens that are about to be retraced.
>
> 2) Will there be information loss during MemVR calculation? As introduced in Q2, highly compressed external information (like the Q-Former of InstructBlip outputs few visual tokens) may lead to poor improvement of performance after retracing. Please ensure that the information from other modalities you'd like to retrace won't be over-compressed.
>
> Once the 2 critical issues listed above can be resolved, we believe that MemVR can be directly and effectively extended to other modalities.
> ﻿
> ﻿
> Please feel free to write your comment if you still have any questions or concerns. Thank you!

---

> > ### Author Response · Authors · 2024-11-22
> > **Summary of response and look forward to the feedback**
> >
> > We greatly appreciate the thoughtful critique and suggestions from Reviewer iT3w. Below is a summary of our revisions and clarifications based on the provided feedback.
> > ﻿
> > - **Addressing Threshold Dependence and Sensitivity:**  We clarified the process of setting MemVR's uncertainty threshold and demonstrated its robustness through experiments. Results showed MemVR’s stable performance across diverse benchmarks even with varying thresholds, confirming that the method is not highly sensitive to this parameter. These analyses highlight MemVR's adaptability to different datasets and architectures.
> > ﻿
> > - **Failure Case Analysis:**  We conducted an in-depth analysis of MemVR's failure cases, particularly on the MME benchmark’s 'Celebrity,' 'Scene,' and 'Landmark' sub-tasks. The failure cases were categorized and attributed to either over-disturbance of the default reasoning process or excessive image complexity. Additionally, we included results showing MemVR's limitations with InstructBLIP due to its highly compressed Q-Former tokens, which lead to information loss. These insights have been incorporated into Section C.3 for a balanced understanding.
> > ﻿
> > - **Improved Presentation Style for Results:**  Responding to feedback on the Q&A format in Section 4.2, we restructured the section into a clear, logical format to improve readability and coherence. This updated presentation ensures that key findings are easily accessible.
> > ﻿
> > - **Correction of Errors in Tables and Data Presentation:**  We meticulously reviewed and corrected all discrepancies in the tables, including incorrect bolding and numerical values, ensuring that the reported results are accurate and consistent throughout the manuscript.
> > ﻿
> > - **Exploring MemVR's Applicability to Other Modalities:**  We discussed how MemVR could be extended to other modalities such as audio or 3D data. We identified critical factors for success, including the need for effective alignment tools and minimal information loss during retracing. This provides a roadmap for future research and adaptations of MemVR to multimodal scenarios.
> > ﻿
> > We hope these revisions and clarifications address your concerns and look forward to any additional feedback or questions.

---

> ### Author Response · Authors · 2024-11-25
> **We are keen to discuss further with you**
>
> Dear Reviewer iT3w,
>
> Thank you for your valuable time and the constructive feedback you have provided. We genuinely hope reviewer iT3w could kindly check our response.
>
> As the deadline for the discussion period is nearing, we would greatly appreciate it if you could kindly let us know whether there are any further questions. Thank you!
>
> Best wishes,
>
> Authors

---

> > ### Author Response · Authors · 2024-11-29
> > **Request for discussion**
> >
> > Dear reviewer iT3w,
> >
> > We have posted some experiment results (e.g., more hyper-parameter experiments and direct analysis), and clarifications to answer your questions. We have also updated our paper according to your suggestions. We would appreciate it if you could let us know whether our responses properly address your concerns.
> >
> > Look forward to your reply.
> >
> > Best regards,
> >
> > Authors

---

### Official Review · Reviewer_eZB5 · 2024-11-04

**Soundness:** 3
**Presentation:** 3
**Contribution:** 2
**Rating:** 6
**Confidence:** 3

**Summary:**

This paper proposes MEMVR, a training-free hallucination mitigation paradigm for MLLMs. MEMVR mimics human patterns of image understanding by revisiting image features to enhance comprehension of image content, utilizing layer-specific uncertainty for dynamic premature layer injection. The authors validate their approach against hallucination benchmarks as well as general-purpose benchmarks, demonstrating its high efficiency.

**Strengths:**

1. The writing quality is high. The paper provides a clear introduction to the hallucination problem in MLLMs, analyzes its causes, presents solutions, and offers a theoretical understanding of the methods used, with clear language and logical coherence.
2. High reusability. Due to the training-free nature of MEMVR, it can be applied in a plug-and-play manner across different MLLMs. The method's simplicity allows for easy adaptation with minimal costs.
3. Sufficient experimentation. The authors validate their approach on both hallucination benchmarks and general-purpose benchmarks, proving its adaptability across various scenarios.

**Weaknesses:**

1. The explanation of the specific process of MEMVR is not sufficiently clear; the discussion of the dynamic injection strategy is brief and potentially confusing.
2. Concerns about usability: The method requires manual tuning of hyperparameters, which may necessitate multiple rounds of adjustments to achieve optimal performance in different scenarios.

**Questions:**

1. In formula (2), the feedforward neural network comprises two fully connected layers. As MEMVR is a training-free method, how are the weight matrices W1 and W2 obtained? If they utilize weights from a pre-trained projector within the MLLM, how can their effectiveness be ensured?
2. In formula (3), K and V are regarded as visual projectors. What is their relationship with the keys and values in the attention mechanism? This seems to differ from the kv cache approach, leading to some ambiguity in the expression.
3. The injection ratio α is a manually tuned hyperparameter. How should the validation set be selected during the tuning process? Does the parameter range differ across scenarios such as healthcare and transportation?
4. In line 279, it is mentioned that details about the dynamic injection strategy will be presented in Appendix C.1; however, C.1 only discusses benchmarks. Due to the unclear introduction of the dynamic injection strategy, I still have some questions. If the authors could provide a detailed explanation of this method, I might consider increasing my score.

---

> ### Author Response · Authors · 2024-11-16
> **Author Response to Reviewer eZB5 (Part 1/2)**
>
> Thank you for your insightful comments and positive evaluation of our work. We address your concerns as follows.
> ﻿
> > Concern 1: The explanation of the specific process of MemVR is not sufficiently clear; the discussion of the dynamic injection strategy is brief and potentially confusing.
> ﻿
>
> Thanks for your feedback. To address your concern, we have provided a clearer explanation in the revised manuscript to ensure that readers understand our methodology.
>
> Here's a detailed explanation of how MemVR is implemented.  In brief, as the title "Look twice before you answer" suggests, this is the main process of our MemVR. Specifically, this involves two issues: 1) at which layer to re-look, and 2) how to re-look at it.
>
> For the first issue - at which layer to re-look, we design a dynamic strategy to select a specific injection layer for 're-look', i.e.,  replenish visual information in the selected layer. In particular, Algorithm 1 in line 277 shows the code flow of our dynamic injection strategy. We first compute the uncertainty of the next token probability on each early layer and set the threshold $\gamma$. When the uncertainty of a specific layer is over $\gamma$, 're-look' is triggered in this layer.
>
> Then, how to 're-look' at it, i.e., how visual retracing is implemented. We take the visual feature after the connector as visual evidence to be re-injected at the trigger layer selected in the previous step. Taking the hidden state $x \in \mathbb{R}^{d}$ of the trigger layer as query, and the visual evidence $z_v\in \mathbb{R}^{n_v \times d}$ as both key and value, then we perform a similar cross-attention operation, i.e, $\operatorname{Retrace}({z}_v \mid {x})=\phi({x} \times {z}_v^\top) \times {z}_v$. Further, in FFN, we re-inject the retrieved visual memory to the hidden-state with ${\operatorname{FFN}}({x} \propto z_v)=(1-\alpha)\operatorname{FFN}(x) + \alpha\operatorname{Retrace}({z}_v \mid {x})$, where $\alpha$ denotes the injection ratio of visual memory.
>
>
> That's the whole process of our method, simple, but efficient and effective. We have revised the specific process of MemVR in the revised manuscript to make our method more clear and accessible.
> ﻿
> > Concern 2: Concerns about usability: The method requires manual tuning of hyperparameters, which may necessitate multiple rounds of adjustments to achieve optimal performance in different scenarios.
>
>
> Thank you for the insightful comment. Yes, MemVR does need a few rounds of adjustments to achieve optimal performance when applying to a brand-new model, this process is typically brief and does not present significant overhead. Take Qwen-VL as an example, we test it with MME and it takes only 6 rounds to find the optimal performance, which is no longer than 2 hours. And we'd like to mention that this does not mean MemVR won't work without manual tuning, as in most cases even a randomly chosen parameter could improve the performance compared to the baseline, indicating a degree of robustness. Besides, MemVR has been tested with 9 different benchmarks, just to ensure that our approach is effective and general, and it turns out that MemVR outperforms all other methods in all evaluation results. More importantly, MemVR is quite easy to extend to other models, such as llava-next series, qwen2 series, etc. with few lines of code, which makes the usability of MemVR pretty impressive.
>
> Here, we supplement the experiments on MME under different injection rates, it can be seen that MemVR is not sensitive to the injection rate setting. Under all injection rates, LLaVA+MemVR can improve the performance compared to the baseline.
>
> | injection rates | 0	| 0.05 | 0.10 | 0.15 | 0.20 | 0.25 | 0.30 |
> | -------- | :--------: | :--------: | :--------: | :--------: | :--------: | :--------: | :--------: |
> | Cognition	| 355.71	| 367.50	| 377.14	| 383.57	| 368.57	| 367.86	| 362.86 |
> | Preception	| 1508.97	| 1514.72	| 1513.70	| 1511.34	| 1512.36	| 1519.13	| 1513.64 |
> | Total Score	| 1864.68	| 1882.22	| 1890.84	| 1894.91	| 1880.93	| 1886.99	| 1876.49 |
>
>
> Thus, hyperparameters do not limit our approach, and we will continue to study how to adjust the injection rate automatically.

---

> ### Author Response · Authors · 2024-11-16
> **Author Response to Reviewer eZB5 (Part 2/2)**
>
> > Q1: In formula (2), the feedforward neural network comprises two fully connected layers. As MemVR is a training-free method, how are the weight matrices W1 and W2 obtained? If they utilize weights from a pre-trained projector within the MLLM, how can their effectiveness be ensured?
>
>
> The W1 and W2 mentioned here are not for visual retracing procedures but for regular FFN calculation. You may refer to our explanation for Concern 1, which may help you fully understand how MemVR is applied.
>
> > Q2: In formula (3), K and V are regarded as visual projectors. What is their relationship with the keys and values in the attention mechanism? This seems to differ from the kv cache approach, leading to some ambiguity in the expression.
>
>
> Sorry for the ambiguity in the paper. Here, we first introduce how visual retracing is implemented. We take the visual feature after the visual projector as visual evidence to be re-injected at the trigger layer. Taking the hidden state $x \in \mathbb{R}^{d}$ of the trigger layer as query, and the visual evidence $z_v\in \mathbb{R}^{n_v \times d}$ as both key and value, then we perform a similar cross-attention operation, i.e, $\operatorname{Retrace}({z}_v \mid {x})=\phi({x} \times {z}_v^\top) \times {z}_v$. Further, in FFN, we re-inject the retrieved visual memory to the hidden-state with ${\operatorname{FFN}}({x} \propto z_v)=(1-\alpha)\operatorname{FFN}(x) + \alpha\operatorname{Retrace}({z}_v \mid {x})$, where $\alpha$ denotes the injection ratio of visual memory. MemVR has nothing to do with kv cache approach. We have provided a clearer explanation in the revised manuscript to ensure that readers understand the details.
>
> > Q3: The injection ratio α is a manually tuned hyperparameter. How should the validation set be selected during the tuning process? Does the parameter range differ across scenarios such as healthcare and transportation?
>
>
> According to the massive experiments we conducted, a proper set of hyperparameters that performs well in one specific benchmark mostly means it also works on most evaluation sets. Say, the best retracing ratio for LLaVA1.5 here, mostly stands for 0.12. Under the retracing ratio set to 0.12, MemVR can improve the performance compared to the baseline on 9 benchmarks across different domains steadily (including code, math, commonsense, lives, etc.).       For Qwen-VL, it would be 0.28.
>
>  When facing different scenarios such as healthcare and transportation,  it needs some changes to the injection ratio, but it is easy to find the optimal parameter. We will continue to study how to adjust the injection rate automatically. You may refer to our explanation for Concern 2.
>
> > Q4: In line 279, it is mentioned that details about the dynamic injection strategy will be presented in Appendix C.1; however, C.1 only discusses benchmarks.
> ﻿
>
> ﻿Sorry, we missed the details in Appendix C.1. The dynamic injection strategy has been explained in Concern 1 and supplemented in the revised version of the paper. These modifications are highlighted in blue in the revised version of our paper to facilitate easy identification.
>
> Please feel free to write your comment if you still have any questions or concerns. Thank you!

---

> ### Author Response · Authors · 2024-11-25
> **We are keen to discuss further with you**
>
> Dear Reviewer eZB5,
>
> Thank you for your valuable time and the constructive feedback you have provided. We genuinely hope reviewer eZB5 could kindly check our response.
>
> As the deadline for the discussion period is nearing, we would greatly appreciate it if you could kindly let us know whether there are any further questions. Thank you!
>
> Best wishes,
>
> Authors

---

> > ### Author Response · Authors · 2024-11-29
> > **Request for discussion**
> >
> > Dear reviewer eZB5,
> >
> > We apologize for reaching out with another reminder of the discussion.
> >
> > Following your suggestions, we have provided additional experimental results and a clearer explanation of our method.
> >
> > We would appreciate your thoughts on whether these updates satisfactorily address your concerns.
> >
> > Best,
> >
> > Authors

---

### Author Response · Authors · 2024-11-25
**General Response (1)**

We appreciate all reviewers’ constructive feedback and positive remarks on the paper.

**To the best of our knowledge, we are the first to mitigate hallucinations and improve general performance in MLLMs with only one regular inference, without incurring added time overhead.** Moreover, MemVR can be extended to **more modalities** instead of being limited to images, and videos. Its high efficiency also allows MemVR to be applied to responsiveness-sensitive domains e.g. healthcare and embodied intelligence.

Here are the characteristics of different methods.
﻿
| Method | Training-free  | Visual Hallucination Mitigation | Generalization | Extended to more modalities | Efficiency | Enhanced Components |
| -------- | -------- | -------- | -------- | -------- | -------- | -------- |
| DoLa (ICLR'24) | ✅ | - | - |  - | ✅ | logits |
| VCD  (CVPR'24) | ✅ | ✅ | -  |  - | - | visual input, logits |
| OPERA (CVPR'24) | ✅ | ✅ | -  |  ✅ |  - | attention matrix |
| HALC [3] (ICML'24) | ✅ | ✅ | -  |  - | - | visual input, logits |
| MVP [4] | ✅ | ✅ | -  |  - | - | visual input, logits |
| VACoDe [5] | ✅ | ✅ | - | -  | - | visual input, logits |
| SID [6] | ✅ | ✅ | -  |  - | - | text input, logits |
| API [1] (ECCV'24) | ✅ | - | ✅  |  - | - | visual input |
| AGLA [2] | ✅ | ✅ | -  | - |  - | visual input, logits |
| **MemVR (ours)** | ✅ | ✅ | ✅ | ✅ |  ✅ | **hidden states** |
﻿

[1] Attention Prompting on Image for Large Vision-Language Models.

[2] AGLA: Mitigating Object Hallucinations in Large Vision-Language Models with Assembly of Global and Local Attention.

[3] HALC: Object Hallucination Reduction via Adaptive Focal-Contrast Decoding, ICML, 2024.

[4] Look, Compare, Decide: Alleviating Hallucination in Large Vision-Language Models via Multi-View Multi-Path Reasoning, 2024.

[5] Vacode: Visual augmented contrastive decoding, 2024.

[6] Self-introspective decoding: Alleviating hallucinations for large vision-language models, 2024.

---

### Author Response · Authors · 2024-12-02
**General Response to All Reviewers**

We are truly grateful for the invaluable time and detailed feedback provided by all the reviewers.

**It is encouraging to see that **Reviewers eZB5, iT3w, tPgc** has recognized the significant contributions and positive aspects of our manuscript**,  such as `writing quality is high`, and `high reusability`, `sufficient experimentation` (Reviewer eZB5);  `MemVR’s approach is innovative`, and `efficiency` (Reviewer iT3w); `good contribution`, and `clearly illustrate the limitations of CD-based methods`, and `clearly presenting`, and `depth analysis` (Reviewer tPgc).

**We have provided detailed responses to each reviewer’s feedback**. In this general response, we outline the major revisions made to our new manuscript based on the valuable suggestions offered by the reviewers. (Please check the changes in blue font in our revision) We hope these responses adequately address any potential concerns from the reviewers, the AC, and SPC.

- **Presentation**: Based on the feedback from **Reviewers eZB5, iT3w, tPgc**, our revisions include ① modifying the method part of Section 3 to more clearly expound the specific process of MemVR, simplify overly complex formulations, and ② updating Section 4.2 on quantitative results to enhance logical presentation and understanding, and ③ revising some figure mistakes on Table 2, 6, 7.

- **Experiments**: According to the feedback from **Reviewers eZB5, iT3w, tPgc**, our revisions include ① adding results and analysis of experiments for hyper-parameters (on the comments and Appendix C) and ② including comparisons and analysis of the bad cases with baseline (see Appendix C), and ③ conducting additional experiments by masking input images with randomly generated noise, offers strong evidence for the effectiveness of the "look twice" mechanism. (on the comments to Reviewer tPgc)

---
**For the feedback from Reviewers YSEL**

Our comments include ① presenting **clear difference and novelty** of our methods (see **General Response (1)** below), ② adding extra experiments on LLaVA-next and InstructBlip requested by Reviewer YSEL (**in fact, we have conducted experiments on 4 different models across 7 benchmarks, more comprehensive than previous methods**), and ③ clarifying a few points regarding all other concerns by Reviewer YSEL, *e.g.*, the phrase "signifies the other side of the coin" at line 114, and the analysis of uncertainty in section 3.2, and concept of “visual prompts”.

We have made sincere efforts to address all the issues raised. **However**, in the last **Reviewer YSEL** claimed that the concept of "visual prompt" lacks context, stating that the writing is **extremely poor** without any other feedback, insisting on the 3 rating.

 **Reviewer YSEL's statement appears to be maliciously biased**.  We hope that the AC and other Reviewers will be able to make a fair judgment. Thank you!

___

We are deeply grateful for the recognition and suggestions from all reviewers. We still respectfully welcome further discussions from all reviewers.

Best regards,

All Authors

---

### Meta-Review · Area_Chair_T5ML · 2024-12-21

**Metareview:**

The paper received mixed feedback, with reviewers acknowledging its innovative "look twice" mechanism for mitigating hallucinations in MLLMs but raising concerns about its clarity and broader impact. The proposed method offers a training-free approach that dynamically injects visual information during inference, which some reviewers found the general idea promising. However, key issues and inconsistencies in presentation/explanation as well as uncertainty measures weakened the overall impression. While the authors addressed many points through revisions and experiments, there is insufficient strong support for its acceptance. The AC recommends reject.

**Additional Comments On Reviewer Discussion:**

The reviewer discussion highlighted both the strengths and shortcomings of the work. While "MemVR"’s efficiency and innovation were recognized, concerns about unclear writing, conceptual inconsistencies, and insufficient evidence for its broader applicability persisted.
Reviewers like tPgc and YSEL expressed concerns about the paper's presentation and did not improve their ratings. Despite attempts to clarify and enhance the manuscript, including additional experiments, the paper’s presentation and framing still left significant gaps. Overall, there was agreement that the work has potential but requires a major rewrite to address these critical issues effectively.

---

### Decision · Program_Chairs · 2025-01-22

Reject